# Synapse-specific opioid modulation of thalamo-cortico-striatal circuits

**William T Birdsong[1†‡*], Bart C Jongbloets[1†], Kim A Engeln[1], Dong Wang[2,3,4], Grégory Scherrer[2,3,4,5], Tianyi Mao[1*]**

[1]Vollum Institute, Oregon Health & Science University, Portland, United States; [2]Department of Anesthesiology Perioperative and Pain Medicine, Stanford Neurosciences Institute, Stanford University, Stanford, United States; [3]Department of Molecular and Cellular Physiology, Stanford Neurosciences Institute, Stanford University, Stanford, United States; [4]Department of Neurosurgery, Stanford Neurosciences Institute, Stanford University, Stanford, United States; [5]New York Stem Cell Foundation – Robertson Investigator, Stanford University, Palo Alto, United States

**\*For correspondence:**
wtbird@med.umich.edu (WTB);
mao@ohsu.edu (TM)

[†]These authors contributed equally to this work

**Present address:** [‡]Department of Pharmacology, University of Michigan, Ann Arbor, United States

**Competing interests:** The authors declare that no competing interests exist.

**Abstract** The medial thalamus (MThal), anterior cingulate cortex (ACC) and striatum play important roles in affective-motivational pain processing and reward learning. Opioids affect both pain and reward through uncharacterized modulation of this circuitry. This study examined opioid actions on glutamate transmission between these brain regions in mouse. Mu-opioid receptor (MOR) agonists potently inhibited MThal inputs without affecting ACC inputs to individual striatal medium spiny neurons (MSNs). MOR activation also inhibited MThal inputs to the pyramidal neurons in the ACC. In contrast, delta-opioid receptor (DOR) agonists disinhibited ACC pyramidal neuron responses to MThal inputs by suppressing local feed-forward GABA signaling from parvalbumin-positive interneurons. As a result, DOR activation in the ACC facilitated poly-synaptic (thalamo-cortico-striatal) excitation of MSNs by MThal inputs. These results suggest that opioid effects on pain and reward may be shaped by the relative selectivity of opioid drugs to the specific circuit components.

DOI: https://doi.org/10.7554/eLife.45146.001

## Introduction

Opioids blunt both sensory-discriminative and affective-motivational dimensions of pain by modulating neuronal activity in the central and peripheral nervous systems (*Oertel et al., 2008*; *Zubieta et al., 2001*; *Corder et al., 2018*). The affective-motivational dimension of pain underlies the aversiveness and negative emotional affect that arise in response to activation of nociceptive inputs (*Gracely, 1992*; *Navratilova and Porreca, 2014*; *Treede et al., 1999*). This study examines how opioids modulate the circuitry involved in affective-motivational pain perception.

In humans, affective pain perception is associated with increased activity in the medial thalamus, as well as the anterior cingulate cortex (ACC) (*Casey et al., 1994*; *Davis et al., 1997*; *Peyron et al., 1999*; *Peyron et al., 2000*). In rodents, chronic pain is associated with hypersensitivity of mediodorsal thalamic neurons to sensory stimuli (*Whitt et al., 2013*) and activation of the ACC has been shown to be aversive using a conditioned place aversion paradigm (*Johansen and Fields, 2004*). In contrast, lesions of the ACC decrease affective-motivational pain responses (*Johansen et al., 2001*). Anatomically, neurons from the medial thalamus (MThal) send glutamate afferents to the cortical regions, including the dorsal and ventral ACC, prefrontal cortex (PFC) and insular cortices (*Hunnicutt et al., 2014*), as well as to the dorsomedial striatum (DMS) (*Hunnicutt et al., 2016*). The ACC in turn projects to the DMS, forming a circuit connecting the MThal and the ACC, both of

which provide convergent glutamate inputs to the DMS. This thalamo-cortico-striatal circuit has been demonstrated to be involved in pain processing, in particular, affective pain perception (*Rainville et al., 1997*; *Price, 2000*; *Fields, 2004*; *Zhang et al., 2015*; *Yokota et al., 2016b*).

Clinically, opioids are used to reduce pain perception by modulating both sensory-discriminative and affective-motivational aspects of pain. Mu-opioid receptors (MORs) and delta-opioid receptors (DORs) are predominantly expressed in the mediodorsal (MD) thalamus and ACC, respectively (*Mansour et al., 1994*; *Erbs et al., 2015*). Injection of opioids into the MD or ACC can relieve pain and induce conditioned place preference in an animal model of chronic pain (*Carr and Bak, 1988*; *Guo et al., 2008*; *Navratilova et al., 2015a*), suggesting a role for opioid modulation of thalamic and cortical circuitry in affective pain.

The striatum is enriched in MORs and DORs, as well as the endogenous opioid ligand enkephalin (*Pert et al., 1976*; *Koshimizu et al., 2008*). Also, opioids have been shown to inhibit glutamate inputs to the striatum, as well as GABA release from local striatal circuitry (*Jiang and North, 1992*; *Hoffman and Lupica, 2001*; *Brundege and Williams, 2002*; *Miura et al., 2007*; *Atwood et al., 2014*; *Banghart et al., 2015*). In this context, the striatum serves as a potentially critical hub for opioid-dependent modulation of fast synaptic transmission in the affective pain circuitry (*Zubieta et al., 2001*).

The current work determines how and where opioids modulate synaptic transmission between the thalamic, cortical and striatal regions that are important for the perception of affective pain. Results revealed opposing roles of the MORs and DORs regarding information flow from the thalamus to the striatum, whereby MOR activation decreased glutamate transmission in the striatum, while DOR activation facilitated glutamate transmission via disinhibition of cortical pyramidal neurons. Thus, MOR and DOR activation are predicted to play opposing roles in pain processing mediated by the thalamo-cortico-striatal circuit. Together, these data identify specific synaptic connections within the thalamo-cortico-striatal circuit that are modulated by opioids and illustrate how different opioid subtypes can independently modulate neuronal communication at the circuit level.

## Results

### Opioid receptor agonists suppress excitatory transmission in the dorsomedial striatum

Because the DMS mediates learning and expression of motivated behaviors in response to both aversive and rewarding stimuli, the opioid sensitivity of glutamate afferents in the DMS was investigated. In acute mouse brain slice preparations, individual striatal medium spiny neurons (MSNs) were identified based on physiological properties (*Kreitzer, 2009*). Whole-cell voltage-clamp recordings were obtained and glutamate release was evoked using electrical stimulation. AMPA receptormediated excitatory postsynaptic currents (EPSCs) were pharmacologically isolated and recorded (*Figure 1—figure supplement 1a–c*). Similar to previously published results in the nucleus accumbens and dorsolateral striatum, application of the mu- and delta-selective opioid agonist [Met$^5$]-enkephalin (ME, 3 µM) significantly decreased the amplitude of the EPSCs. This inhibition was reversed upon washout of ME (*Figure 1—figure supplement 1c*; ME: 77.8 ± 4.6% of baseline; washout: 90.5 ± 2.6% of baseline; baseline vs ME: W(11) = 1, p<0.01) (*Jiang and North, 1992*; *Hoffman and Lupica, 2001*; *Brundege and Williams, 2002*). The AMPA receptor antagonist NBQX (3 µM) eliminated the evoked currents (*Figure 1—figure supplements 1c* and 4.0 ± 2.6% of baseline; baseline vs NBQX: W(5) = 0, p<0.05).

### Mu-opioid receptor agonists suppress thalamic but not cortical glutamatergic inputs to the dorsomedial striatum

Anatomic mapping in mice indicated that glutamate inputs from the medial thalamus and ACC converged in the DMS (*Figure 1a–b* and *Figure 1—figure supplement 1d–e*) (*Hunnicutt et al., 2016*). In addition, MORs and DORs appear to be enriched in the medial thalamus and midline cortical structures (including the ACC), respectively (*Erbs et al., 2015*; *Wang et al., 2018*). To test whether anatomically distinct expression of MORs and DORs in the thalamus and the cortex confer specific opioid sensitivity to these two inputs to the striatum, an optogenetic approach was used to isolate

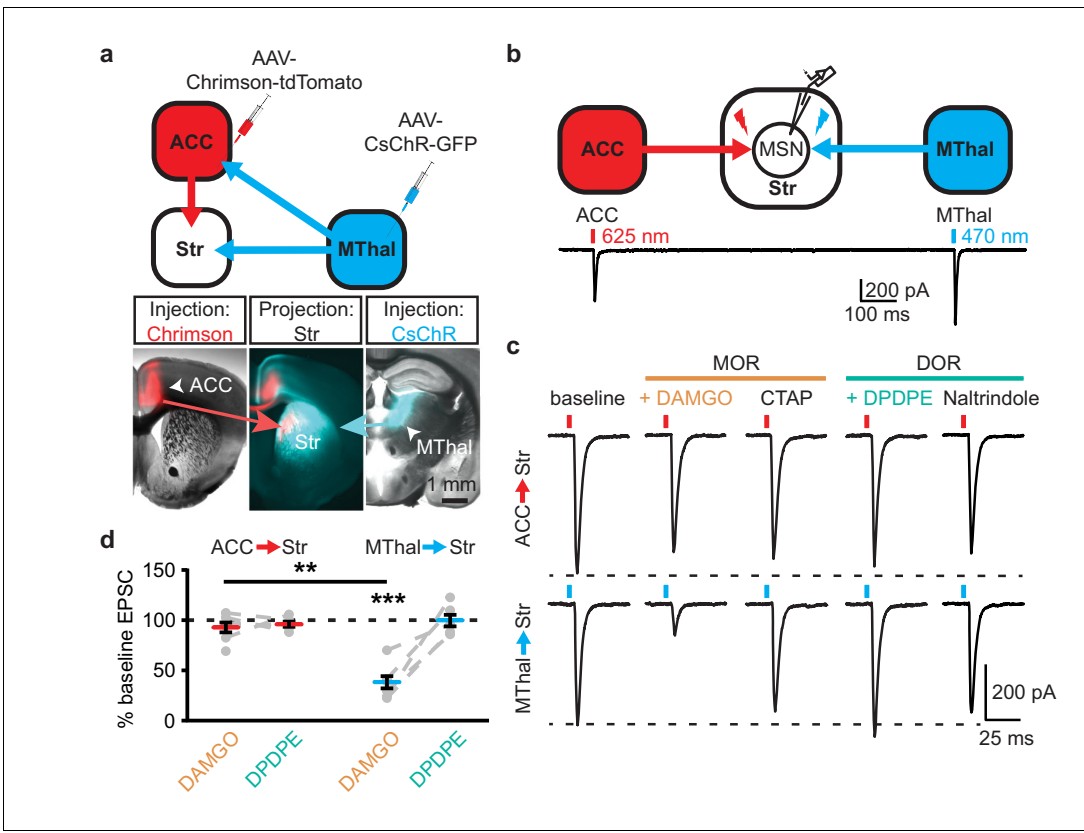

**Figure 1.** Mu-opioid agonists suppress thalamic but not cortical inputs to single MSNs in the striatum. (**a**) Schematic (upper panel) and an acute mouse brain slice example (lower panel) of viral injection design and the axonal projections to the striatum, respectively. Overlaid brightfield and epifluorescent images showing the injection site of Chrimson (left image, red) in the ACC, and CsChR (right image, cyan) in the MThal, and convergent axonal projections from both injections to the DMS (center image). (**b**) Schematic (upper panel) and representative recordings (lower panel) for optical excitation. (**c**) Example oEPSCs of individual MSNs evoked by 625 nm (from the ACC, upper traces), and by 470 nm (from the MThal, lower traces) light pulses. The MOR (orange label) agonist DAMGO (1 μM) was perfused followed by the MOR antagonist CTAP (1 μM). Following CTAP, the DOR (teal label) agonist DPDPE (1 μM) was perfused followed by the moderately-selective DOR antagonist naltrindole (0.3 μM). Red bars: 3 ms of 625 nm light stimulation; blue bars: 1 ms of 470 nm light stimulation. (**d**) Summary data of dual wavelength excitation of the ACC and MThal input oEPSCs recorded from single MSNs. Data are plotted as the percentage of baseline current following exposure to DAMGO or DPDPE for inputs from the ACC and MThal (N = 5, n = 8, Linear mixed model: 3-way interaction, opioid type (mu vs. delta opioid) x input source (ACC vs. MThal) x drug condition (baseline vs. agonist vs. antagonist), F (4,8)=2.938, p=0.091; MThal$_{baseline \times}$ $_{DAMGO}$: $z = 4.738$, p<0.001; MThal$_{baseline \times DAMGO}$ vs. ACC$_{baseline \times DAMGO}$; $z = -3.026$, p<0.01). Mean ± standard error of the mean. Str: striatum.

DOI: https://doi.org/10.7554/eLife.45146.002

The following figure supplements are available for figure 1:

**Figure supplement 1.** Mu-opioid agonists suppress thalamic, but not cortical inputs.
DOI: https://doi.org/10.7554/eLife.45146.003

**Figure supplement 2.** Single channelrhodopsin injections reproduced specific effect of mu-opioid-mediated inhibition of thalamic but not cortical inputs.
DOI: https://doi.org/10.7554/eLife.45146.004

the specific thalamic and cortical inputs onto MSNs in the DMS. Recombinant adeno-associated viruses (AAVs) encoding two optically-separable channelrhodopsin variants were injected into the MThal and ACC of three- to five-week-old mice (*Figure 1a*). The blue light-sensitive channelrhodop-sins (CsChR or ChR2(H134R)) were expressed in the MThal while the red light-sensitive channelrho-dopsin (Chrimson) was expressed in the ACC (*Nagel et al., 2005*; *Klapoetke et al., 2014*). Expression of one channelrhodopsin variant alone demonstrated wavelength-selectivity of optically-

evoked excitatory postsynaptic currents (oEPSCs) in response to brief pulses of either blue (470 nm) or red (625 nm) light, and minimal cross-contamination from undesired light stimulation was observed under these conditions (*Figure 1—figure supplement 1d–e*). Following co-expression of Chrimson in the ACC and either CsChR or ChR2(H134R) in the MThal, illumination with both blue (470 nm, MThal inputs) and red light (625 nm, ACC inputs) evoked robust oEPSCs in individual MSNs in the DMS (*Figure 1b*). The MOR-selective agonist DAMGO inhibited oEPSC amplitude to MThal stimulation (470 nm light) in a reversible manner, and did not alter oEPSC amplitude to ACC stimulation (*Figure 1c*; DAMGO $I_{MThal}$: 38.2 ± 6.1% of baseline, $z$ = 4.738, p<0.001; $I_{ACC}$: 92.8 ± 4.9%, $z$ = 0.459, p=0.647). These observations demonstrated both the optical separation of the thalamic and cortical inputs and the ability of MOR agonists to selectively inhibit excitatory MThal inputs. Despite the apparent expression of DORs in the midline cortical regions, DOR activation by its selective agonist DPDPE did not inhibit the ACC, or MThal inputs (*Figure 1b–d*; DPDPE $I_{MThal}$: 99.6 ± 5.7% of baseline, $z$ = 0.106, p=0.916; $I_{ACC}$: 95.6 ± 2.6%, $z$ = 0.196, p=0.845).

Similar results were obtained when Chrimson and CsChR were injected into the MThal and the PFC, respectively (*Figure 1—figure supplement 1f–g*). DAMGO decreased oEPSC amplitude from the MThal inputs while it had no effect on oEPSCs from the PFC inputs (*Figure 1—figure supplement 1h–i*; DAMGO $I_{MThal}$: 22.2 ± 3.7% of baseline, $z$ = 3.497, p<0.001; $I_{PFC}$: 93.9 ± 5.6%, $z$ = 0.052, p=0.958). DPDPE produced no significant changes in oEPSC amplitude from either input (DPDPE $I_{MThal}$: 96.5 ± 3.9% of baseline, $z$ = 0.087, p=0.931; $I_{PFC}$: 81.6 ± 3.9%, $z$ = 0.672, p=0.502). These results indicate that MThal glutamate inputs onto MSNs were preferentially inhibited by activation of MORs, while DOR agonists had little to no modulatory effect on the MThal, ACC, or PFC inputs to the MSNs in the DMS. These findings of mu- and delta-opioid specificity were further confirmed with similar experiments using single channelrhodopsin variant viral injections into the ACC, PFC, or MThal (*Figure 1—figure supplement 2*).

To ensure that these opioid-sensitive inputs indeed originated from the MThal, *Slc17a6-cre* mice, which express Cre-recombinase in vGlut2-positive cells (*Vong et al., 2011*), were injected with DIO-ChR2(H134R) virus which restricted the expression of ChR2 only to the thalamus (*Wu et al., 2015*). Similar to the results from wild-type mice, activation of MORs by DAMGO, but not DORs by DPDPE, resulted in inhibition of the MThal inputs to the MSNs (*Figure 2*; $I_{DAMGO}$: 18.1 ± 3.9% of baseline, W (6) = 0, p<0.05; $I_{DPDPE}$: 100.5 ± 7.9%, W(5) = 7, p=1). Furthermore, MOR agonists also inhibited inputs from the anterior medial thalamus, suggesting the general effects of opioid inhibition of thalamic inputs to the DMS (*Figure 1—figure supplement 2*).

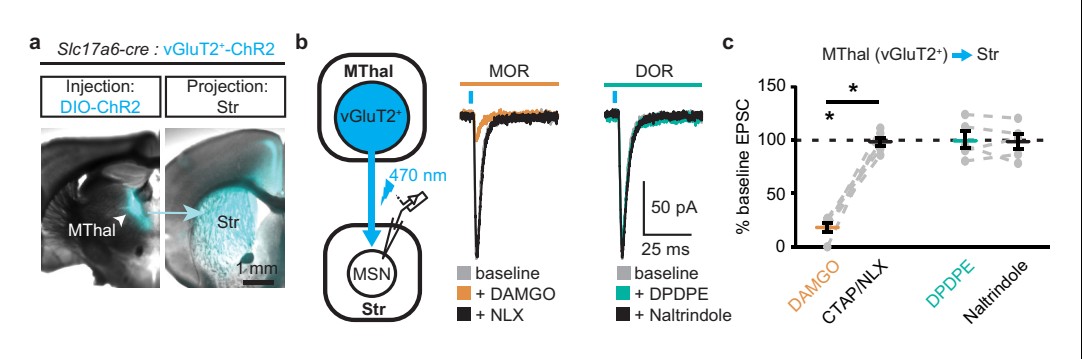

**Figure 2.** Mu-opioid agonists selectively suppress thalamic inputs from vGluT2-positive thalamic neurons. (**a**) Example overlaid brightfield and epifluorescent images showing Cre-dependent expression of ChR2(H134R)-EYFP (cyan) in the MThal (left panel) and axonal projections into the DMS (right panel) following injection of AAV-DIO-ChR2(H134R)-EYFP in the MThal of *Slc17a6-Cre* mice, which express Cre-recombinase in vGluT2-positive cells. (**b**) Experimental schematic showing optical stimulation of glutamate inputs in the DMS (left panel). Representative traces of oEPSCs showing effects of MOR agonist DAMGO (1 μM, middle panel, orange) and antagonist naloxone (NLX, 1 μM, middle panel, black), and the DOR agonist DPDPE (1 μM, right panel, teal) and DOR antagonist naltrindole (0.3 μM, right panel, black). Blue bars: 1 ms of 470 nm light stimulation. (**c**) Summary data of oEPSCs showing effects of MOR agonist DAMGO and antagonist CTAP or NLX (1 μM), and the DOR agonist DPDPE and DOR antagonist naltrindole. DAMGO/(CTAP/NLX): N = 3, n = 6, *SM* = 9.33, p<0.01; DPDPE/naltrindole, N = 3, n = 5, *SM* = 0, p=1.0. Skillings-Mack test followed by paired Wilcoxon signed-ranks test *post-hoc* analysis. Mean ± standard error of the mean. *p≤0.05; *SM*: Skillings-Mack statistic; Str: striatum.
DOI: https://doi.org/10.7554/eLife.45146.005

## Thalamostriatal and thalamocortical projections can arise from the same medial thalamic neuronal population

Single neuron tracing in the rat has demonstrated that medial thalamic neurons can send collaterals to both the cortex and striatum (*Otake and Nakamura, 1998*; *Kuramoto et al., 2017*). To determine whether the opioid-sensitive population of thalamic neurons in mouse project to both the ACC and DMS, two approaches were used. First, red and green fluorescent retrograde transported beads (retrobeads) were injected into the ACC and DMS, respectively (*Figure 3—figure supplement 1b*). The injection sites were then localized based on the mouse brain atlas (*Franklin and Paxinos, 2001*), *Figure 3—figure supplement 1b–c*). Retrogradely labeled somas were found in the lateral MD and the central lateral (CL) thalamus (*Figure 3—figure supplement 1c–g*). In brain sections containing the MD thalamus (selected figures 44, 45, and 46 of the Franklin and Paxinos atlas, second edition; *Figure 3—figure supplement 1d*), there was a substantial fraction of neurons that projected to the DMS that also contained retrograde beads originating from the ACC (21 ± 4% of striatal-projecting neurons colocalized with ACC-projecting neurons). Second, to further determine whether cortical-projecting thalamic neurons send collateral axons to the striatum, a retrogradely transported virus, rAAV-retro (*Tervo et al., 2016*), encoding Cre-recombinase was injected into the ACC, along with AAV-GFP which served to indicate the injection site of rAAV2-retro-Cre virus and to visualize cortico-striatal axons (*Madisen et al., 2015*; *Tervo et al., 2016*). In the same mice, Cre-dependent AAV-FLEX-TdTomato virus was injected into the MThal (*Figure 3a–d*). TdTomato would be expected to be expressed only in the thalamic neurons projecting to the ACC area that were also infected by rAAV-retro-Cre virus. Indeed, TdTomato-positive neurons were found in the MThal (*Figure 3b*). GFP expression in the ACC indicated the rAAV-retro virus injection site (*Figure 3c*), and TdTomato-expressing axon terminals, originating from thalamic neurons, were also visible in the ACC (*Figure 3c*). Prominent GFP-expressing axons originating from the ACC and TdTomato-expressing axon collaterals originating from the MThal were observed in the DMS (*Figure 3d*). These results indicate the existence of thalamic neurons that project to both the ACC and DMS, and further, that ACC-striatal and MThal-striatal projections can innervate anatomically overlapping areas in the DMS (*Hunnicutt et al., 2016*).

To confirm that the apparent axon collaterals in the DMS form functional synapses rather than passing through the DMS en route to the ACC, rAAV-retro-Cre was injected into the ACC and a Cre-dependent ChR2-expressing virus AAV-DIO-ChR2(H134R)-EYFP was injected into the MThal, conferring the ability within the striatum to optogenetically activate potential thalamic axons originating from ACC-projecting MThal neurons (*Figure 3e*). Optical illumination evoked glutamate-mediated EPSCs in striatal MSNs that were potently inhibited by ME (*Figure 3f*; $I_{ME}$: 22.4 ± 6.6% of baseline, N = 3, n = 5, paired t-test, p<0.01). These data suggest the existence of an opioid-sensitive thalamic neuronal population projecting to both the DMS and ACC.

## Mu-opioid agonists suppress excitatory thalamic inputs to the ACC, while delta-opioid agonists suppress feed-forward inhibition

Since the excitatory MThal inputs to the DMS were strongly inhibited by MOR activation and at least a fraction of those thalamic neurons project to both the ACC and DMS, we hypothesized that the excitatory MThal inputs to the ACC would be inhibited by MORs. Inputs from the MD thalamus have been reported to synapse directly onto layer 2/3 (L2/3) and 5 (L5) pyramidal neurons, and also to trigger GABA release onto pyramidal neurons via innervation of L2/3 and L5 parvalbumin-positive (PV) interneurons in the ACC (*Delevich et al., 2015*). Both optically evoked EPSCs and feed-forward inhibitory postsynaptic currents (oIPSCs) were measured in L2/3 and L5 pyramidal neurons of the ACC (*Figure 4* and *Figure 4—figure supplement 1*; also see Materials and methods). Similar to the results from recordings of the MSNs in the DMS, oEPSCs of the L2/3 and L5 pyramidal neurons in the ACC from the MThal inputs were potently inhibited by the MOR agonist DAMGO but unaffected by the DOR agonist DPDPE, suggesting expression of MORs but not DORs on thalamic glutamate terminals in the ACC (*Figure 4b–c*, and *Figure 4—figure supplement 1b–c*; $I_{DAMGO}$: 54.0 ± 5.0% of baseline, W(17) = 1, p<0.001; $I_{DPDPE}$: 93.4 ± 7.0%, W(14) = 36, p=0.312; L2/3; $I_{DAMGO}$: 69.3 ± 3.3% of baseline, W(15) = 0, p<0.001; $I_{DPDPE}$: 92.9 ± 5.4%, W(9) = 9, p=0.129). These oEPSCs were of thalamic origin since track injection of ChR2 dorsal to the thalamus resulted in only sporadic and insignificant oEPSCs in the ACC (*Figure 4—figure supplement 2*).

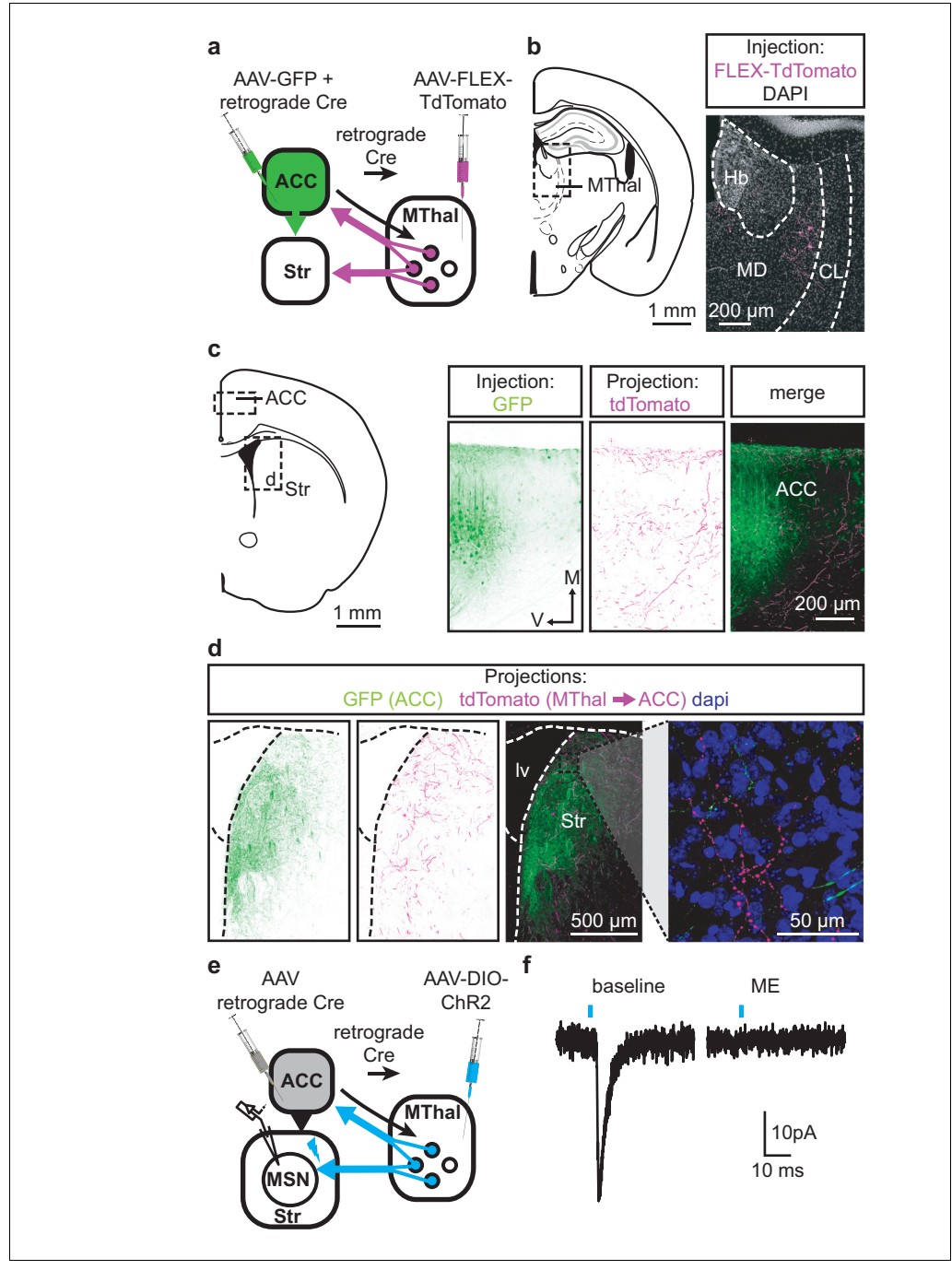

**Figure 3.** Individual mediodorsal thalamic neurons project to both the DMS and ACC. (a) Schematic of the rAAV-retro-Cre and Cre-dependent AAV-DIO-TdTomato injections. (b) Fluorescent image of cell bodies expressing TdTomato following injections shown in (a) in the MThal (magenta, right panel), with corresponding mouse brain atlas section (left panel). (c) Representative mouse brain atlas section showing approximate origin of images taken from ACC (left panels) and DMS as shown in (d). Images of the ACC showing AAV-GFP injection site and axons from the MThal (magenta). (d) Images of the DMS showing overlapping axons from both the ACC (green) and MThal (magenta). Rightmost panel shows high magnification image taken at the black box demarcation in the left panel. Cell nuclei are stained with DAPI (blue). (e) Schematic of retrograde rAAV-retro-Cre and Cre-dependent AAV-DIO-ChR2(H134R)-EYFP injections, and recordings of MSNs in the DMS. (f) An example trace of oEPSCs of a MSN in the DMS from a mouse injected as shown in (e). ME: opioid agonist [Met[5]]-enkephalin. Blue bars: 1 ms of 470 nm light stimulation. V: ventral; M: midline; Hb: habenula; CL: centrolateral thalamus; MD: mediodorsal thalamus; Str: striatum; lv: lateral ventricle. Mouse brain atlas sections from *Franklin and Paxinos (2001)*.

*Figure 3 continued on next page*

*Figure 3 continued*

DOI: https://doi.org/10.7554/eLife.45146.006

The following figure supplement is available for figure 3:

**Figure supplement 1.** A subset of mediodorsal thalamic neurons send collaterals to both the ACC and DMS.
DOI: https://doi.org/10.7554/eLife.45146.007

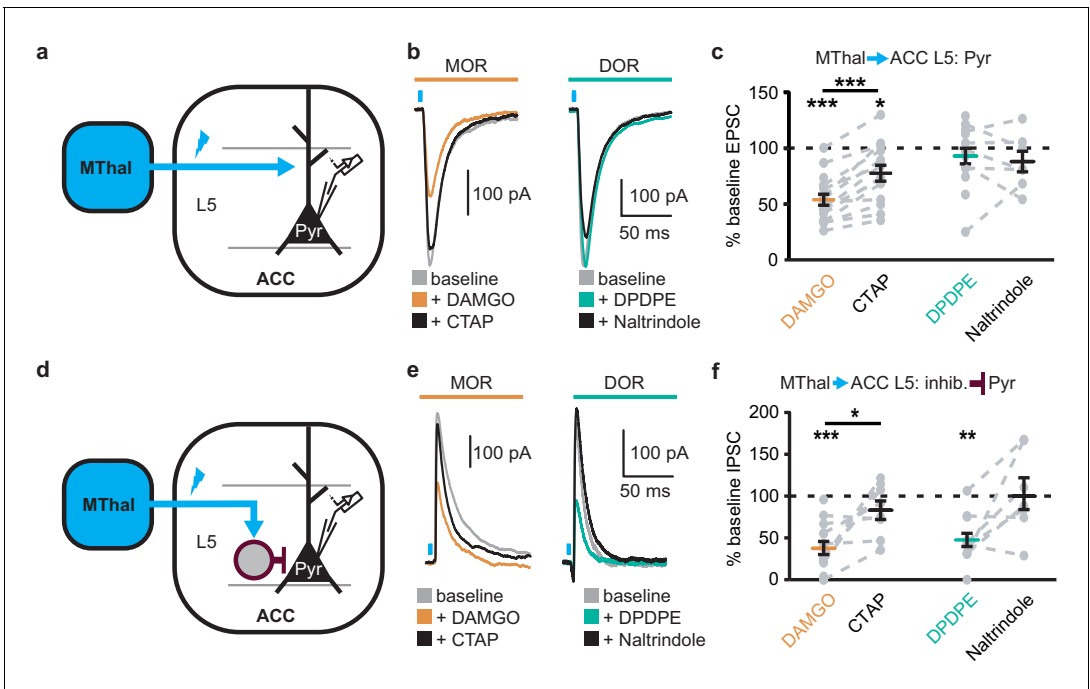

**Figure 4.** Mu-opioid agonists suppress thalamic inputs to pyramidal neurons in the ACC while delta-opioid agonists suppress cortical feed-forward inhibition in the ACC. (a–c) oEPSCs of the pyramidal neurons in the ACC elicited by optical stimulation of MThal input. (a) Schematic of ChR2 injection, MThal optical stimulation, and recording of oEPSCs of the layer 5 (L5) pyramidal neurons (Pyr) in the ACC. Blue: ChR2 expression and optical stimulation. (b) Example traces of oEPSCs elicited from optical stimulation of the MThal terminals in the ACC during baseline (gray), application of DAMGO (1 µM, left panel, orange), followed by CTAP (1 µM, left panel, black), or application of DPDPE (1 µM, right panel, teal,) followed by naltrindole (0.3 µM, right panel, black). Blue bars: 1 ms of 470 nm light stimulation. (c) Summary data of oEPSCs of all recording as shown in (b) with responses plotted as a percent of the baseline. DAMGO: N = 13, n = 17; CTAP: N = 12, n = 14; SM = 22.85, p<0.001; DPDPE: N = 11, n = 15; naltrindole: N = 5, n = 7, SM = 0.989, p=0.610. (d–f) oIPSCs of the pyramidal neurons from the MThal optical stimulation, via inhibition through interneurons in the ACC. (d) Schematic of ChR2 injection, MThal optical stimulation, and recording of the pyramidal neurons in the ACC via feed-forward inhibition. Blue: ChR2 expression and optical stimulation; magenta: outline of an interneuron. (e) Example traces of oIPSCs elicited from optical stimulation of the MThal terminals in the ACC during baseline (gray), application of DAMGO (1 µM, left panel, orange) followed by CTAP (1 µM, left panel, black), or application of DPDPE (1 µM, right panel, teal) followed by naltrindole (0.3 µM, right panel, black). (f) Summary data of oIPSCs for all recordings as in (b) with responses plotted as a percent of the baseline. DAMGO: N = 9, n = 14; CTAP, N = 7, n = 8, SM = 15.68, p<0.001; DPDPE: N = 6, n = 12; naltrindole: N = 5, n = 7, SM = 7.426, p<0.05. Skillings-Mack test followed by paired Wilcoxon signed-ranks test *post-hoc* analysis. *p≤0.05; **p≤0.01; ***p≤0.001. Mean ± standard error of the mean. *SM*: Skillings-Mack statistic; Blue bars: 1 ms of 470 nm light stimulation.

DOI: https://doi.org/10.7554/eLife.45146.008

The following figure supplements are available for figure 4:

**Figure supplement 1.** Opioid inhibition of synaptic currents onto layer 2/3 pyramidal neurons in the ACC.
DOI: https://doi.org/10.7554/eLife.45146.009

**Figure supplement 2.** Verification of origins of optically-evoked response.
DOI: https://doi.org/10.7554/eLife.45146.010

**Figure supplement 3.** Opioid agonists induced suppression of inhibitory synaptic currents onto layer 5 pyramidal neurons in the ACC is mediated via delta-opioid receptors.
DOI:

Surprisingly, both DAMGO and DPDPE potently inhibited GABA-mediated oIPSCs from the MThal to the ACC pyramidal neurons (*Figure 4e–f* and *Figure 4—figure supplement 1e–f*; $I_{DAMGO}$, 37.9 ± 7.8% of baseline, W(14) = 0, p< 0.001; $I_{DPDPE}$, 47.6 ± 7.9%, W(11) = 1, p<0.01; $I_{DAMGO}$, 61.2 ± 9.5% of baseline, W(12) = 1, p<0.001; $I_{DPDPE}$, 62.7 ± 5.4%, W(10) = 0, p<0.01). To confirm the DPDPE-mediated oIPSC inhibition (*Figure 4e-f*) is indeed modulated by DOR activation, we pre-treated the brain slices with naltrindole. This pretreatment occluded the DPDPE-mediated inhibition of oIPSCs (*Figure 4—figure supplement 3*). Since PV-positive neurons reportedly contribute to feed-forward inhibition of MThal inputs to L5 pyramidal neurons in the ACC (*Delevich et al., 2015*), opioid modulation of oIPSCs of PV neurons onto L5 pyramidal neurons in the ACC was measured. *Pvalb-cre⁺/⁻;Ai32⁺/⁻* mice (*Madisen et al., 2015*; *Hippenmeyer et al., 2005*) were used to express ChR2(H134R) in PV neurons, and oIPSCs were recorded from L5 pyramidal neurons (*Figure 5a*). The oIPSCs were potently inhibited by DPDPE but not DAMGO (*Figure 5b–c*; $I_{DAMGO}$: 104.1 ± 6.1% of baseline, W(11) = 25, p=0.998; $I_{DPDPE}$: 44.9 ± 5.7%, W(15) = 0, p<0.001), suggesting that DORs were expressed on PV neurons in the ACC.

Colocalization of DORs and parvalbumin was investigated at mRNA and protein levels (*Figure 5d–f*). Images of fluorescent RNA probes for endogenous DORs (*Oprd1*) and parvalbumin (*Pvalb*) revealed that the majority of *Pvalb*-positive cells were also *Oprd1*-positive (*Figure 5d,f*; 88.7 ± 1.4% of *Pvalb*-positive cells expressed detectable levels of *Oprd1*). A knockin mouse line expressing GFP-fused DORs (*Scherrer et al., 2006*) was used to probe for the presence of DOR proteins in sections stained for parvalbumin proteins. Similarly, 95.0 ± 1.2% of parvalbumin protein-positive neurons expressed detectable levels of GFP (*Figure 5e–f*). Together, these data indicate that DORs are

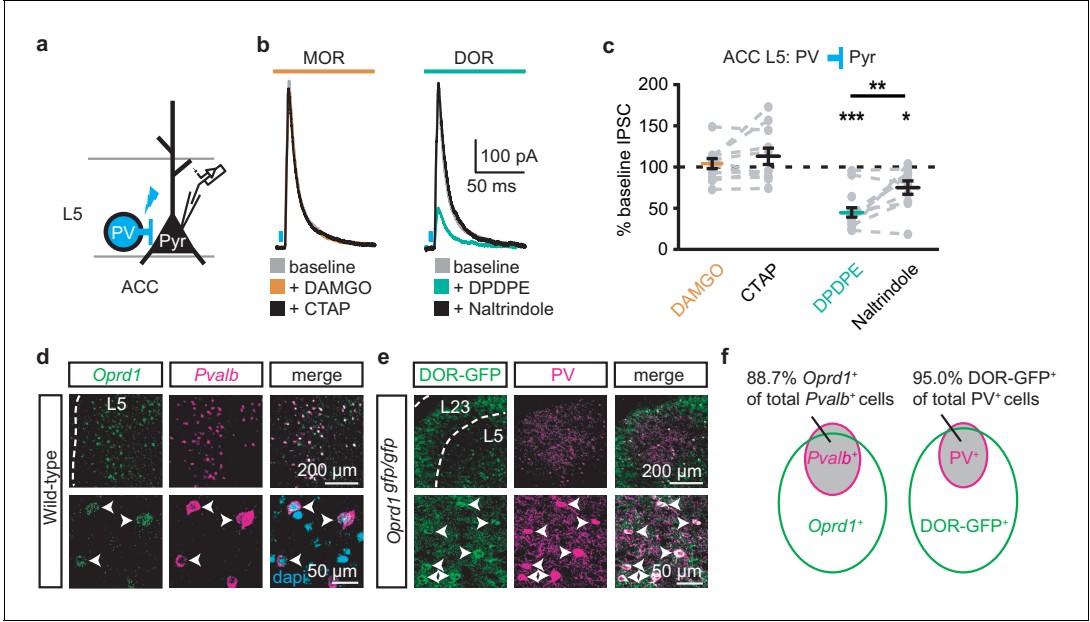

**Figure 5.** DORs expressed on PV-positive interneurons suppress oIPSCs onto layer five pyramidal neurons. (a) Schematic of ChR2 expression and recording of oIPSCs of parvalbumin-positive interneurons (PV) to layer 5 (L5) pyramidal neurons (Pyr) in the ACC of *Pvalb-cre⁺/⁻;Ai32⁺/⁻* mice. Blue: ChR2 expression and optical stimulation. (b) Example traces of oIPSCs during baseline (gray), application of DAMGO (1 µM, left panel, orange) and followed by CTAP (1 µM, left panel, black), or application of DPDPE (1 µM, right panel, teal) followed by naltrindole (0.3 µM, right panel, black). Blue bars: 1 ms of 470 nm light stimulation. (c) Summary data of PV interneurons to pyramidal neuron oIPSCs for all recording as in (b). Responses plotted as a percent of the baseline. DAMGO/CTAP: N = 5, n = 11, *SM* = 0.005, p=0.998; DPDPE: N = 8, n = 15; naltrindole: N = 5, n = 10, *SM* = 19.60, p<0.001. (d) In-situ hybridization in wild-type mouse brain sections containing the ACC stained with probes against mRNA coding for *Oprd1* (*Oprd1*, left panel, green) and parvalbumin (*Pvalb*, middle panel, magenta) and overlaid with DAPI (right panel, cyan). (e) Immunohistochemistry of brain sections of the ACC from a *DOR-GFP* knockin mouse and stained with anti-GFP antibodies (left panel, green), and anti-parvalbumin antibodies (middle panel, magenta), and overlaid (right panel). (f) Venn diagram quantifying overlap of *Oprd1*-positive and *Pvalb*-positive (left panel, N = 2, n = 8), and DOR-GFP-positive and PV-positive cells in the mouse ACC (right panel, N = 2, n = 15). Skillings-Mack test followed by paired Wilcoxon signed-ranks test *post-hoc* analysis. *, p<0.05; **, p<0.01; ***p<0.001. Mean ± standard error of the mean. *SM*: Skillings-Mack statistic.

DOI: https://doi.org/10.7554/eLife.45146.011

expressed on PV neurons, consistent with their role in inhibition of GABA release onto pyramidal neurons.

## Delta-opioid agonists increase cortical excitability

As DOR agonists selectively reduced feed-forward inhibition of the local PV neurons to pyramidal neurons in the ACC, activation of DORs was predicted to disinhibit ACC pyramidal neurons resulting in increased cortical excitability. Optical excitation of MThal terminals in the ACC evoked action potential (AP) firing in ACC L5 pyramidal neurons measured using a cell-attached recording configuration, recorded as action currents in voltage-clamp mode (*Figure 6a–c*). When APs were elicited in approximately 50% of the trials, activation of MORs by DAMGO decreased the fraction of trials that evoked action potentials (0.4 ± 0.1 fold change relative to baseline, paired t-test, p<0.01), while activation of DORs by DPDPE led to a significant increase in AP firing probability (5 ± 3.1 fold change relative to baseline, paired t-test, p<0.05). This was consistent with an inhibitory effect of MOR activation and a disinhibitory effect of DOR activation on pyramidal neuron firing rates (*Figure 6b–c*) with no significant effect on action potential latency (*Figure 6—figure supplement 1*). Thus, DOR activation in the ACC resulted in disinhibition of the pyramidal neurons within the ACC, which led to increased cortical excitability.

## Delta-opioid agonists facilitated thalamo-cortico-striatal signaling

The increased excitability upon DOR activation in L5 pyramidal neurons in the ACC was hypothesized to propagate to MSNs in the DMS via corticostriatal projections and result in increased glutamate release in the striatum. To test this hypothesis, AAV-ChR2(H134R) was injected into the MThal. Brain slices containing both the ACC and DMS were prepared (*Figure 7a*). Simultaneous recordings were made from L5 pyramidal neurons in the ACC (voltage recording) and MSNs in the DMS (current recording). Laser illumination was used to focally excite axons within the ACC or the DMS. When the ACC was illuminated, optogenetically stimulated MThal axons triggered action potentials in the L5 pyramidal neurons in the ACC, which then propagated to the DMS, and in turn, triggered a polysynaptic oEPSCs in the MSNs (*Figure 7a*). In contrast, focal illumination in the DMS resulted in a

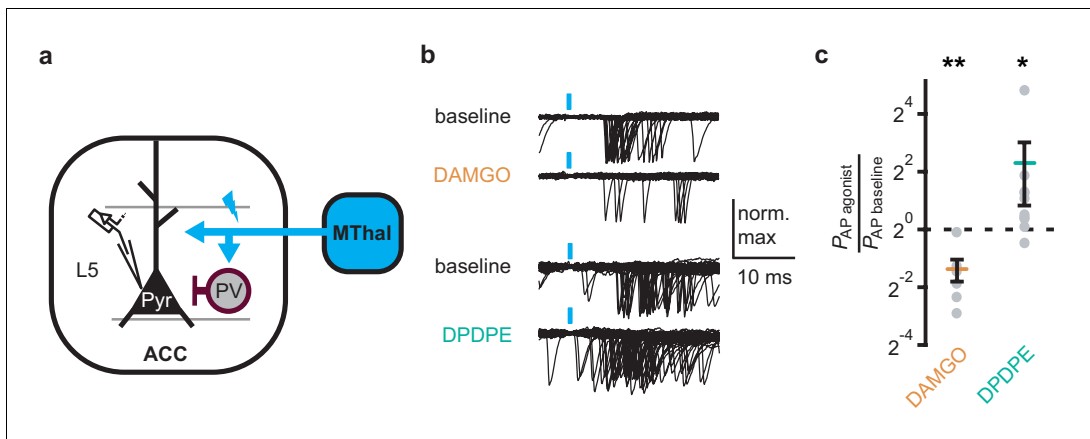

**Figure 6.** DOR activation results in increased cortical excitability. (**a**) Schematic of ChR2 expression and recording of optically-evoked action potentials (APs, recorded as action currents using voltage-clamp mode) using a loose cell-attached recording configuration. Blue: ChR2 expression and optical stimulation; magenta: outline of a parvalbumin-positive interneuron (PV). (**b**) Example traces of 50 trails from a single layer 5 (L5) pyramidal neuron (Pyr) in which APs were evoked by optical stimulation (blue bars) under baseline conditions, or in the presence of DAMGO (1 μM, orange), or DPDPE (1 μM, teal). (**c**) Summary data plotted on a log2 scale for action potential firing probability ($P_{AP}$) in the presence of drugs ($P_{AP\ agonist}$) relative to baseline ($P_{AP\ baseline}$). Paired t-test; *p≤0.05; **p≤0.01. DAMGO: N = 3, n = 7; DPDPE: N = 5, n = 8.
DOI: https://doi.org/10.7554/eLife.45146.012

The following figure supplement is available for figure 6:

**Figure supplement 1.** Latencies of thalamocortical-evoked action potentials in ACC pyramidal neurons.
DOI: https://doi.org/10.7554/eLife.45146.013

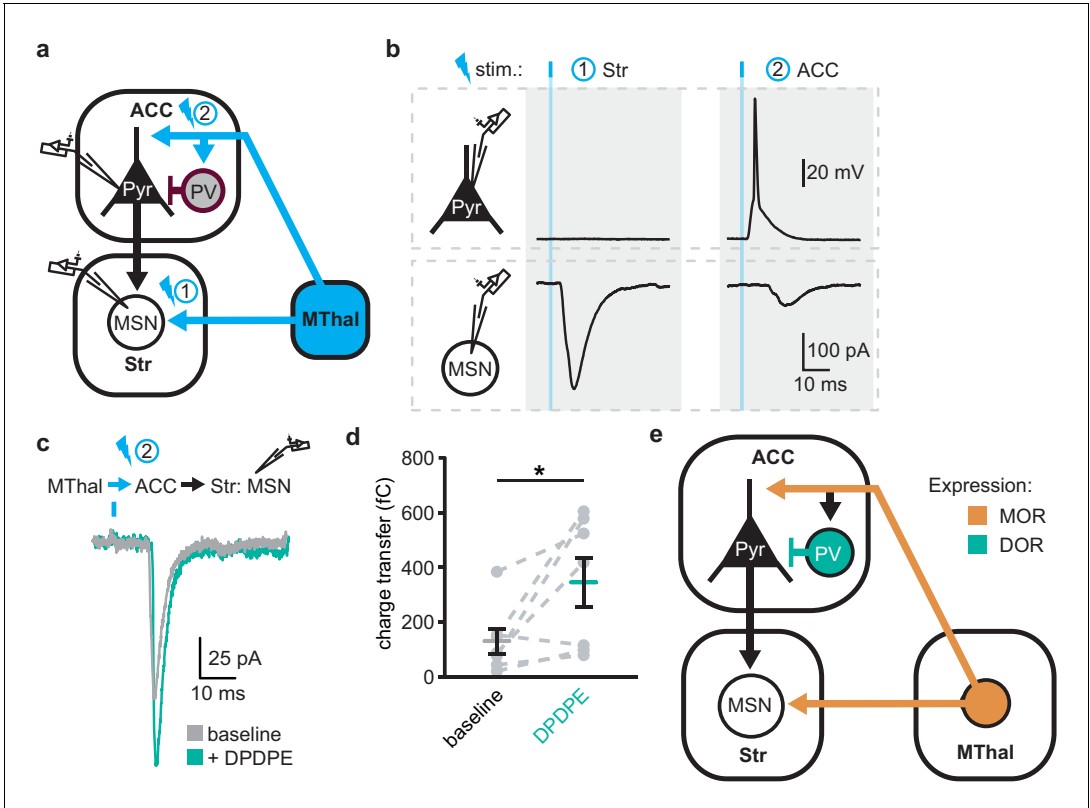

**Figure 7.** DOR activation disinhibits thalamo-cortico-striatal circuits. (**a**) Schematic of ChR2 expression and dual recordings in layer 5 (L5) pyramidal neurons (Pyr) in the ACC and MSNs in the DMS. Blue: ChR2 expression and optical stimulation; magenta: outline of a parvalbumin-positive interneuron (PV). 470 nm light stimulation locations are shown as 1 and 2. (**b**) Example traces of current-clamp recording from a L5 pyramidal neuron in the ACC (upper panels), and voltage-clamp recording from a MSN in the DMS (lower panels) in response to light stimulations at location 1 (Str) and location 2 (ACC). (**c**) Example traces of poly-synaptic current evoked by optical stimulation of MThal terminals in the ACC (location 2) while recording from an MSN in the DMS during baseline (gray), and in presence of DPDPE (1 μm, teal). (**d**) Summary data of the charge transfer of poly-synaptic oEPSCs in the MSNs evoked by optical stimulation of location 2 in the ACC. Paired Wilcoxon signed-ranks test. *p≤0.05; Gray: baseline; teal: DPDPE. (**e**) Summary schematic depicting a part of the affective and motivational pain circuit consisting of the MThal, ACC and DMS. Str: striatum.

DOI: https://doi.org/10.7554/eLife.45146.014

The following figure supplement is available for figure 7:

**Figure supplement 1.** EPSCs and action potential latencies within the thalamo-cortico-striatal circuits.

DOI: https://doi.org/10.7554/eLife.45146.015

short latency oEPSC in the MSNs in the DMS with no measurable responses in the L5 pyramidal neurons in the ACC. These results indicated that MThal-ACC-DMS circuits were preserved in this slice preparation. Long-latency poly-synaptic oEPSCs were also confirmed in MSNs in the DMS following widefield optical stimulation of the ACC (*Figure 7b* and *Figure 7—figure supplement 1a*). Application of DPDPE increased the charge transfer of the poly-synaptic MThal→ACC→DMS oEPSCs (*Figure 7b–c*; $Q_{baseline}$: 129.4 ± −46.2 fC; $Q_{DPDPE}$: 344.7 ± 90.5 fC, W(7) = 2, p<0.05). Amplitude changes of the poly-synaptic oEPSC in response to DPDPE, however did not reach statistical significance (*Figure 7—figure supplement 1b*; $I_{DPDPE}$: 196.1 ± 41.0%, W(7) = 3, p=0.145). Taken together, the results indicate the presence of both MORs and DORs in the thalamo-cortico-striatal circuitry, where DORs primarily facilitated information flow in the indirect pathway from the MThal via the ACC to the DMS, while MORs suppressed information flow from the MThal directly to the DMS.

## Discussion

Despite the importance of the thalamo-cortico-striatal loop in affective pain and reward, the exact circuit mechanisms and how they are modulated by opioids are not fully understood. Previously, we

described the mesoscopic anatomical connectivity between subregions within the thalamus and cortex, and the convergence of their axonal projections within the striatum (*Hunnicutt et al., 2016*). Here, guided by such information, we used both anatomical and functional approaches to demonstrate the connectivity of the MThal, ACC and DMS. We found that projections from the MThal and ACC converged in the DMS, rather than the ventral striatum which has been well studied for reward and drug addiction. Further, we found different opioid receptors differentially affect projection-specific synapses (*Figure 7e*). Specifically, activation of MORs suppressed both MThal-ACC and MThal-DMS excitatory synapses, however, activation of DORs enhanced the excitatory input from the MThal to the pyramidal neurons in the ACC by disinhibiting local feed-forward inhibition mediated by PV interneurons in the ACC. This DOR-mediated disinhibition of MThal-ACC synapses has functional significance at the network level in that it facilitates information to flow from the MThal to the DMS via the ACC. Our results suggests that opioid effects on pain and reward are shaped by the relative selectivity of opioid drugs to the specific circuit components.

## Convergence of thalamic and cortical inputs to the striatum

Glutamate afferents from the midline cortical structures and medial thalamus converge to the DMS in the affective pain pathway. Here, we demonstrated that single MSNs in the DMS can receive inputs from both the midline cortex and medial thalamus. These results establish the anatomical basis for investigating convergence and integration within this circuit. Being able to detect convergence of MThal and ACC inputs at the level of the individual cell also paves the way for future studies of this circuit with single cell resolution at the population level.

## Individual MThal neurons projects to both the ACC and DMS

Consistent with studies demonstrating cortical and striatal projections for midline thalamic neurons in rats (*Otake and Nakamura, 1998*; *Kuramoto et al., 2017*), the simultaneous retrograde bead injections localized to the rostromedial ACC and DMS demonstrated double-labeling of individual neurons in the lateral MD and CL thalamus, suggesting that in mice single thalamic neurons can project to both the cortex and striatum. It should be acknowledged that the exact location of the retrograde bead labeled neurons relative to defined borders of the thalamic nuclei is not perfectly accurate since a small range of slices containing the MThal were fit to a single plane of the mouse atlas (*Franklin and Paxinos, 2001*) without correcting for irregularities such as brain slice angles. Furthermore, the diffusion of retrograde beads in brain tissue is limited such that each injection is relatively small and localized, thus, the number of colocalized neurons is not an absolute measure of the fraction of MD neurons projecting to both the ACC and DMS. Despite these constraints, the retrograde bead injections and retrograde viral labeling experiments both suggested the presence of a population of neurons in the MThal that project to both the ACC and DMS. We also confirmed that these collaterals originating from the MThal made functional en passant synapses in the DMS. Based on these tracing experiments and convergence of the cortical and thalamic inputs to the striatum experiments mentioned above, a circuit with a direct monosynaptic arm from the MThal to the DMS, and an indirect poly-synaptic arm from the MThal via ACC to the DMS is described.

## Opioid modulation of the thalamo-cortico-striatal circuitry

The MOR agonist DAMGO had a strong inhibitory effect on the MThal inputs to the DMS, whereas neither DAMGO nor the DOR agonist DPDPE had a significant effect on the ACC inputs to the DMS. These results indicate that the midline thalamostriatal pathway stands as a major source of opioid-regulated glutamate inputs to the DMS, and that MOR agonists alter the relative influence of cortical and thalamic inputs on DMS excitability. The MThal inputs to the ACC were also inhibited by MOR agonists. Given the extensive MOR-sensitive axonal projections from the MThal to the ACC, it is possible that the analgesic and rewarding effect of morphine injection into the ACC is due in part to inhibition of thalamic glutamate inputs to the ACC (*Navratilova et al., 2015b*). Together, MOR activation may selectively suppress information from MThal to the striatum either directly or indirectly, but may leave intact other information that is also going through the ACC to the striatum. This may contribute to the selective analgesic and rewarding effect of morphine without affecting other cognitive functions.

There are two major subtypes of MSNs in the striatum, dopamine receptor 1 (D1)-positive and dopamine receptor 2 (D2)-positive MSNs each making up ~50% of total MSNs. While we have not observed any bi-modal distribution or obvious heterogeneity in MSN responses, it will be interesting to investigate the cell type-specific and input-specific opioid sensitivity in this circuit in the future.

## Opposing effects of opioid subtypes on circuitry modulation

Although all opioid receptors couple to the inhibitory $G_i$ pathway, their effect on the circuit can be distinct. In this study, the lack of strong inhibition of DORs on glutamate transmission in the cortico-striatal projections together with the potent DOR inhibition of GABA release from PV interneurons to pyramidal neurons in the ACC allows for DORs to function in a disinhibitory manner. DOR activation effectively leads to hyper-excitable ACC circuits, while MOR activation functions in an inhibitory manner, dampening glutamate release in the thalamo-cortico-striatal circuit. These results complement a previous study in which DOR activation resulted in increased activity in the rat insular cortex following dental pulp stimulation, while MOR activation decreased the insular cortex activity (*Yokota et al., 2016a*). A follow-up study found that inhibitory transmission from fast-spiking interneurons to pyramidal neurons was inhibited by DOR activation (*Yokota et al., 2016b*). The current results extend these findings to the mouse ACC and identify that specifically, PV interneurons are a target of DOR inhibition.

While PV interneurons have been suggested to selectively mediate feed-forward inhibition of thalamocortical transmission in the ACC, it is important to note that the current findings do not rule out potential influence of opioids on other interneuron populations. In fact, immunohistochemical and in situ hybridization data presented here suggest that DOR expression is not restricted to PV interneurons, as only approximately 20% of DOR-positive neurons were parvalbumin-positive (*Figure 4e*). Furthermore, certain neurons in the ACC have been shown to express MORs (*Tanaka and North, 1994*; *Vogt et al., 1995*; *Wang et al., 2018*), suggesting that there are additional opioid-sensitive neurons within the local ACC circuits.

It has recently been shown that MOR agonists can modestly inhibit insular-striatal glutamate transmission (*Muñoz et al., 2018*). In this study ACC-striatal transmission was relatively insensitive to MOR and DOR agonists, while PFC-striatal transmission appeared marginally sensitive to the DOR agonist DPDPE. Thus, there may be heterogeneity in the opioid sensitivity of cortical projections to the striatum. The current results also suggest that MORs and DORs are positioned to serve opposing functions in regulating affective pain circuitry and reward and motivated behaviors. In addition, an important aspect is how endogenous opioid receptor ligands modulate this circuit. Future technological development might allow the investigation of endogenous ligand release and their effects on specific components of the thalamo-cortico-striatal loop, as outlined in this work.

The MD, ACC, and striatum have been demonstrated to play roles in pain- and reward-related behaviors (*Kawagoe et al., 2007*; *LaLumiere and Kalivas, 2008*; *Le Merrer et al., 2009*; *Harte et al., 2011*; *Navratilova et al., 2012*). Therefore, some aspects of the analgesic and rewarding/addictive effects of opioids may arise from modulation within the poly-synaptic circuits between the MThal, ACC and striatum described here. Increased ACC activity is associated with increased affective pain intensity and decreased ACC activity is associated with analgesia (*Johansen and Fields, 2004*). The current results suggest that drugs acting in the ACC at either MOR or DOR have the potential to modulate this circuitry. These data also suggest that commonly prescribed and abused opioids, which primarily act through MOR, could alter the relative influences of glutamate inputs to the striatum in addition to their postsynaptic effects on striatal MSNs.

## Materials and methods

### Key resources table

| Reagent type (species) or resource | Designation | Source or reference | Identifiers | Additional information |
|---|---|---|---|---|
| Strain, strain background (*M. musculus, C57BL/6J*) | wildtype | Jackson Laboratories | Stock # 000664 RRID: IMSR_JAX: 000664 | |

*Continued on next page*

*Continued*

| Reagent type (species) or resource | Designation | Source or reference | Identifiers | Additional information |
|---|---|---|---|---|
| Genetic reagent (M. musculus) | Ai32 (B6;129S-Gt (ROSA)26Sortm 32(CAG-COP4*H13 4R/EYFP)Hze/J) | Jackson Laboratories | Stock # 012569 RRID: IMSR_JAX: 012569 | PMID: 22446880 |
| Genetic reagent (M. musculus) | Ai9 (B6.Cg-Gt(ROSA)26Sor tm9(CAG-TdTomato)Hze/J) | Jackson Laboratories | Stock # 007909 RRID: IMSR_JAX:007909 | PMID: 20023653 |
| Genetic reagent (M. musculus) | Pvalb-IRES-Cre (B6.129P2-Pvalbtm 1(cre)Arbr/J) | Jackson Laboratories | Stock # 008069 RRID: IMSR_JAX:007909 | PMID: 15836427 |
| Genetic reagent (M. musculus) | Slc17a6-IRES-Cre (B6.Slc17a6tm2 (cre)Lowl/J) | Jackson Laboratories | Stock # 016963 RRID: IMSR_JAX: 016963 | PMID: 21745644 |
| Genetic reagent (Dependoparvovirus) | AAV2-syn-hChR2 (H134R)-EYFP | UNC virus vector core | NA | |
| Genetic reagent (Dependoparvovirus) | AAV2/1-CAG-hChR2 (H134R)-TdTomato | UPenn Vector Core | Addgene plasmid # 28017 | PMID: 21982373 |
| Genetic reagent (Dependoparvovirus) | AAV2-syn-ChrimsonR-TdTomato (*Klapoetke et al., 2014*) | UNC virus vector core | NA | PMID: 24509633 |
| Genetic reagent (Dependoparvovirus) | AAV2-syn-CsChR-GFP (*Klapoetke et al., 2014*) | UNC virus vector core | NA | PMID: 24509633 |
| Genetic reagent (Dependoparvovirus) | AAVrg-pmSyn1-EBFP-cre (*Madisen et al., 2015*; *Tervo et al., 2016*) | Addgene | Cat# 51507 | PMID: 25741722, 27720486 |
| Genetic reagent (Dependoparvovirus) | AAV2-FLEX-CAG-TdTomato | UNC virus vector core | NA | |
| Genetic reagent (Dependoparvovirus) | AAV2-SSpEMBOL-Chicken-beta-actin (CBA)-GFP | UNC virus vector core | NA | |
| Genetic reagent (Dependoparvovirus) | AAV1-CAG-hChR2 (H134R)-TdTomato | UPenn Vector Core | Addgene plasmid # 28017 | PMID: 21982373 |
| Genetic reagent (Dependoparvovirus) | AAV2-DIO-EF 1alpha-ChR2 (H134R)-EYFP | UNC virus vector core | NA | |
| Antibody | Living Colors DsRed Polyclonal Anibody | Takara Bio/ Clontech | Cat# 632496 RRID: AB_10013483 | dilution: 1:500 |
| Antibody | Anti-GFP chicken Polyclonal IgY fraction | Thermo Fisher | Cat# A10262 RRID: AB_2534023 | dilution: 1:500 |
| Antibody | Alexa Fluor 488 goat anti-chicken Polyclonal IgG | Thermo Fisher | Cat# A11039 RRID: AB_2534096 | dilution: 1:750 |
| Antibody | Alexa Fluor 594 goat anti-rabbit Polyclonal IgG (H + L) | Thermo Fisher | Cat# A11012 RRID: AB_2534079 | dilution: 1:750 |
| Antibody | Anti-parvalbumin goat Polyclonal | Swant | Cat# PVG 214 RRID: AB_10000345 | dilution: 1:1000 |
| Antibody | Anit-GFP chicken Polyclonal | Abcam | Cat# ab13970 RRID: AB_300798 | dilution: 1:1000 |
| Chemical compound, drug | Mecamylamine | R and D Systems/Tocris | Cat# 2843 | |
| Chemical compound, drug | Scopolamine | Sigma Aldrich | Cat# S1013 | |

*Continued on next page*

*Continued*

| Reagent type (species) or resource | Designation | Source or reference | Identifiers | Additional information |
|---|---|---|---|---|
| Chemical compound, drug | SR95531 | Hello Bio | Cat# HB0901 | |
| Chemical compound, drug | Picrotoxin | Hello Bio | Cat# HB0506 | |
| Chemical compound, drug | [Met$^5$]-enkephalin | Sigma Aldrich | Cat# M6638 | |
| Chemical compound, drug | DAMGO | Sigma Aldrich | Cat# E7384 | |
| Chemical compound, drug | CTAP | R and D Systems/Tocris | Cat# 1560 | |
| Chemical compound, drug | Naloxone | Abcam | Cat# ab120074 | |
| Chemical compound, drug | DPDPE | Sigma Aldrich | Cat# E-3888 | |
| Chemical compound, drug | Naltrindole | Sigma Aldrich | Cat# N-115 | |
| Chemical compound, drug | ICI 174,864 | R and D Systems/Tocris | Cat# 0820 | |
| Chemical compound, drug | MK801 | Hello Bio | Cat# HB0004 | |
| Chemical compound, drug | CPP | R and D Systems/Tocris | Cat# 0173 | |
| Chemical compound, drug | CGP 55845 | R and D Systems/Tocris | Cat# 1248 | |
| Chemical compound, drug | Bestatin | Sigma Aldrich | Cat# B8385 | |
| Chemical compound, drug | Thiorphan | Sigma Aldrich | Cat# T6031 | |
| Chemical compound, drug | MPEP | R and D Systems/Tocris | Cat# 1212 | |
| Chemical compound, drug | Red Retrobeads IX | Lumafluor Inc | Cat# R180 | |
| Chemical compound, drug | Green Retrobeads IX | Lumafluor Inc | Cat# R180 | |
| Chemical compound, drug | DNQX | Sigma Aldrich | Cat# D0540 | |
| Chemical compound, drug | QX314 | R and D Systems/Tocris | Cat# 2313 | |
| Commercial assay, kit | RNAscope Multiplex Fluorescent Assay | Advanced Cell Diagnostics | Cat# 320850 | |
| Commercial assay, kit | RNAscope Probe-Mm-Oprd1 | Advanced Cell Diagnostics | Cat# 427371 | |
| Commercial assay, kit | RNAscope Probe-Mm-Pvalb-C2 | Advanced Cell Diagnostics | Cat# 421931-C2 | |
| Software, algorithm | FIJI 1.49b | Wayne Rasband, NIH | PMID: 22743772 | |
| Software, algorithm | AxoGraph 1.4.4 | Axograph | | |
| Software, algorithm | Microsoft Excel 2011 | Microsoft Corp. | | |
| Software, algorithm | Illustrator CS5 | Adobe Systems | | |
| Software, algorithm | Photoshop CS5 | Adobe Systems | | |

*Continued on next page*

*Continued*

| Reagent type (species) or resource | Designation | Source or reference | Identifiers | Additional information |
|---|---|---|---|---|
| Software, algorithm | Prism 6 | GraphPad | | |
| Software, algorithm | Imaris | Bitplane | | |
| Software, algorithm | Zen | Zeiss | | |
| Software, algorithm | Chart 5 | AD Instruments | | |
| Software, algorithm | Matlab r2007b and r2018a | Mathworks | | |
| Software, algorithm | R (3.5.0) | R Development Core Team | | |
| Software, algorithm | Ephus | Vidrio Technologies, LLC | | |
| Software, algorithm | Custom data analysis software | https://gitlab.com/maolab/opi_syn_circuit.git | | Data and analysis scripts related to quantification in manuscript and rebuttal |

All procedures were approved by Oregon Health and Science University Institutional Animal Care and Use Committee (IACUC) under protocol IP00000955, and Institutional Biosafety Committee under protocol IBC-10–40. Mice of both sexes were used in all experiments and were five to eight weeks of age at the time of brain slice preparation. Stereotaxic injections were performed on three- to five-week-old mice. Mice were housed in group housing, given free access to food and water, and maintained on a 12 hr light/dark cycle. List of resources can be found in Key resources table. The software, and data sets generated and analyzed during the current study are available upon request to the corresponding authors.

## Viral injection

Stereotaxic injections were performed as previously described (*Hunnicutt et al., 2014*; *Birdsong et al., 2015*) to deliver recombinant adeno-associated virus (AAV) to express channelrhodopsin variants. Briefly, mice were deeply anesthetized with isofluorane and head fixed into a stereotaxic alignment system (Kopf Instruments, Tujunga, CA, with custom modifications). Small holes were drilled through the skull above the desired injection site and a glass pipette filled with virus was lowered through the hole to the desired injection depth. A small volume (20–40 nl) of virus was injected (WTB and KAE: Nanoject II, Drummond Scientific, Broomall, PA; BCJ: custom-built injector based on a MO-10, Narishige, Amityville, NY). Injection coordinates are listed below in mm for medial/lateral (M/L), anterior/posterior from bregma (A/P), and dorsal/ventral from the top of the skull directly over the target area. Target areas included (in mm): medial thalamus (MThal): M/L: 0.55, A/P: −1.2, D/V: 3.6; anteromedial thalamus (AMThal): M/L: 0.55, A/P: −0.4, D/V: 3.4; anterior cingulate cortex (ACC): M/L: 0.4, A/P: 0.7, D/V:1.6; prefrontal cortex (PFC): M/L: 0.45, A/P: 1.75, D/V: 1.6, dorsomedial striatum (DMS); M/L: 1.5; A/P: 0.55, D/V: 3.6 to 3.3.

## Brain slice electrophysiology

Two to three weeks after viral injection, acute brain slices were prepared. Two different slicing protocols were used depending on whether recordings were being obtained from the striatum or ACC. Recordings were made from both slicing solutions and similar results were obtained.

For striatal recordings, coronal brain slices (250–300 µm) were prepared from either ice-cold or room temperature Krebs buffer containing (in mM): 125 NaCl, 21.4 NaHCO$_3$, 11.1 D-glucose, 2.5 KCl, 1.2 MgCl$_2$, 2.4 CaCl$_2$, 1.2 NaH$_2$PO$_4$,~305 mOsm, supplemented with 5 µM MK-801 and saturated with 95% O$_2$/5% CO$_2$. Slices were incubated in oxygenated Krebs buffer supplemented with 10 µM MK-801 for 30 min at 33°C and then maintained in a holding chamber at 22–24°C.

For cortical recordings, coronal brain slices (300–350 µm) were prepared in a carbogen saturated choline-based cutting solution containing (in mM): 110 choline chloride, 25 NaHCO$_3$, 25 D-glucose, 2.5 KCl, 7 MgCl$_2$, 0.5 CaCl$_2$, 1.25 NaH$_2$PO$_4$, 11.5 sodium ascorbate, and three sodium pyruvate,~315

mOsm. Slices were incubated in oxygenated artificial cerebrospinal fluid (aCSF) containing (in mM): 127 NaCl, 25 NaHCO₃, 25 D-glucose, 2.5 KCl, 1 MgCl₂, 2 CaCl₂, and 1.25 NaH₂PO₄ for 30 min at 33–34°C and then maintained in a holding chamber at 22–24°C.

Two experimenters (WTB and BCJ) using two rigs performed whole-cell recordings; experimenters' initials below note differences between experimental setups. There were no differences in results between experimenters so all data were pooled. Recordings were obtained at near-physiological temperature (32–34°C) from slices superfused with (BCJ) oxygenated aCSF supplemented with (in µM, see Key resources table): 10 GABA_B-receptor antagonist CGP 52432, 10 GABA_A-receptor antagonist SR-95531, 10 nicotinic acetylcholine receptor mecamylamine, 10 muscarinic acetylcholine receptor antagonist scopolamine, 0.3 metabotropic glutamate receptor five antagonist MPEP, 5 NMDA receptor antagonist CPP, or (WTB) oxygenated Krebs supplemented with (in µM, see Key resources table): 0.2 GABA_B-receptor antagonist CGP 55845, 10 GABA_A-receptor antagonist picrotoxin, one mecamylamine, 0.1 muscarinic acetylcholine receptor antagonist atropine or 0.1 scopolamine and 0.3 MPEP, preincubated in 5 MK-801.

## Electrophysiology data acquisition

Borosilicate pipettes (2.8–4 MΩ; Sutter Instruments, Novato, CA) were filled with potassium gluconate-based internal solution (in mM: 110 potassium gluconate, 10 KCl, 15 NaCl, 1.5 MgCl₂, 10 HEPES, 1 EGTA, 2 Na₂ATP, 0.3 Na₂GTP, 7.8 phosphocreatine; pH 7.35–7.40;~280 mOsm) for striatal recordings. Putative MSNs were identified by their morphology and stereotypic physiological properties. Evoked excitatory postsynaptic currents (EPSCs) were recorded in whole-cell voltage-clamp mode at −75 mV holding potential.

To facilitate measurement of both GABA_A- and AMPA-mediated currents, cortical recordings were obtained using a low-chloride cesium gluconate solution (in mM: 135 Glucaronic acid, 1 EGTA, 1.5 MgCl₂, 10 HEPES, 2 Na2ATP, 0.3 GTP, 7.8 Na₂ Phosphocreatine, titrated to pH 7.35–7.4 with CsOH,~280 mOsm and 3 QX314 chloride added fresh before experiment) in the absence of GABA_A antagonists. Sodium and chloride reversal-potentials were empirically determined by recording spontaneous EPSCs/IPSCs, isolated by presence of GABAergic or glutamatergic antagonists, respectively, under a range of holding potentials (−90 to 40 mV with 5 mV increments each for 1 min recording duration). oEPSC and oIPSC were recorded at −55 and 5 mV, respectively.

Whole-cell voltage and current clamp recordings were collected by WTB using an Axopatch 200A amplifier (Molecular Devices, San Jose, CA), digitized at 20 kHz (Instrutech ITC-16, New York, NY), and recorded (Axograph X software), and by BCJ using an Multiclamp 700B amplifier (Molecular Devices) digitized at 10 kHz and recorded with *Ephus* software (www.ephus.org). Optically evoked currents were elicited by LED illumination through the microscope objective (WTB and BCJ, Olympus, Tokyo, Japan, BX51W with 60X, 1.0 NA water immersion objective, except for polysynaptic circuit activation which utilized a 20X, 0.5 NA water immersion objective), or by laser illumination (473 nm Crystal Laser, Reno, NV) through the microscope objective (BCJ, details see *Mao et al., 2011*) Neuron). In brief, laser beam position was controlled by galvanometer scanners (Cambridge Technology, Bedford, MA). Beam was passed through an air objective (4 x, 0.16 NA, UPlanApo, Olympus, beam diameter ~8–16 µm). Timing and light power of the laser stimulation was controlled by a TTL-controlled shutter (Uniblitz, Rochester, NY) with typical dwell time 1 ms (up to 5 ms for polysynaptic and loose cell-attached experiments in *Figures 6* and *7*), and a circular gradient neutral-density filter of 0.04–1.5 optical density (Edmund Optics, Barrington, NJ) set to yield typical 3–5 mW power after objective, respectively. For LED stimulation a TTL-controlled LED driver and 470 nm LED (Thorlabs, Newton, NJ) were used to illuminate the slice directly over the recorded cell generally with ~1 mW of power for 0.5 ms or 1 ms, although power was increased or decreased if evoked currents were unusually weak or strong, respectively. For two-wavelength optical excitation, single flashes of 470 nm (one msec,<0.5 mW) and 625 nm LEDs (3 ms, 1.4 mW, Thorlabs) were used.

## Electrophysiology data analysis

Data collected by WTB were analyzed either in Axograph or Matlab, data collected by BCJ in *Ephus* were analyzed in Matlab. Pooled data were processed in Matlab and R. All collected data were analyzed using the same protocols. Peak current amplitude was calculated relative to mean current during 50 ms baseline prior to the stimulus. In three cases the recorded oIPSCs (recorded as positive

current) was completely blocked in the presences of DAMGO, yielding a small negative current deflection (residual EPSC-mediated sodium current as a result of imperfect voltage-clamp at the estimated sodium-reversal potential). To calculate agonist effects on the oIPSCs relative to baseline recordings the small negative currents were substituted with a positive current value (0.00001 pA). Signal latencies were calculated between stimulation and 10% of peak current for oEPSCs or peak of action potentials. Charge transfer was calculated by integrating recorded current during a defined time window following photostimulation, as described below, and was corrected by baseline charge transfer during a time window measured immediately prior to stimulation. In *Figure 4—figure supplement 2* a large portion of 'non-responding' cells were observed in experiments from 'tail' injected mice (hippocampus +retrosplenial cortex). In order to determine the charge transfer in an unbiased way, we first determined the averaged onset, rise, and decay time of monosynaptic inputs to the recorded cells (thalamic and cortical inputs to the MSNs, thalamic inputs to the L2/3 or L5 pyramidal neurons), and determined the integration time window (averaged rise +decay time) and the time position relative to stimulation (averaged onset time) for individual cell types. These cell type-specific integration time windows and positions allowed us to determine the charge transfer in an unbiased way in both responding and non-responding cells. Charge transfer of polysynaptic currents (*Figure 7c*) were calculated as integrated current between onset and decay time window, both defined as 10% of peak current.

## Retrograde bead injection and tissue processing

Red and green retrobeads (Lumafluor, Durham, NC) were injected similar to viral injections described above with the exception that 100 and 200 nl of retrobeads were injected into the cortex and striatum, respectively. Five days post-injection, mice were transcardially perfused with phosphate-buffered saline (PBS) followed by 4% formaldehyde in PBS. Brains were dissected and stored overnight in 4% formaldehyde in PBS. Sections containing injection sites were sliced at 100 µm thickness, and sections containing the thalamus were sliced at 50 µm thickness and stained with DAPI (300 nM in PBS) to label nuclei.

## Fluorescent imaging and analysis for retrograde bead labeling

Thalamic sections containing red and green retrobeads were imaged at 20x magnification using three laser lines (405, 488 and 561 nm) on a Yokogawa spinning disk confocal microscope (Carl Zeiss, Oberkochen, Germany). A Z-stack series was acquired with a 0.44 µm optical section thickness, and for each image 3 x 3 tiles with 15% overlap were applied. Laser power and exposure times were identical for all images. Retrobead-labeled cells were manually counted on Imaris 9.0 software (Bitplane) and each slice was recounted twice with consistent exposure settings to generate an average. A cell was counted if at least two punctate fluorescent spots were orthogonally detected directly adjacent to DAPI-labeled nuclei. Colocalization of dual-color spots was quantified using a built-in Imaris function with criteria that red and green-labeled spots must be spaced no more than 8 µm apart.

Subsequent imaging alignment was performed according to the mouse brain atlas (*Franklin and Paxinos, 2001*), and aligned using Adobe Photoshop and Illustrator. Sections containing the mediodorsal thalamus were aligned to the corresponding mouse brain atlas section (Figure 45, *Franklin and Paxinos, 2001*) using the hippocampus, habenula, and third ventricle as landmarks; and red, green, and red/green colocalized cells were hand-traced onto the aligned atlas figure. Injection sites were imaged on a brightfield epifluorescent macroscope (Olympus MVX10) using identical settings, and uniform thresholds were established using fluorescent values from the top 90% brightest pixels (ImageJ). The outline of each injection site was aligned to a representative mouse brain atlas section to view the average injection area (for the ACC, Figure 22 and for the DMS, Figure 28 of the Franklin and Paxinos atlas, second edition; *Franklin and Paxinos, 2001*).

## Retrograde viral cell labeling

A 60 nl viral mix containing a 3:1 ratio of rAAV2-retro-Cre (AAVrg-pmSyn1-EBFP-cre, Addgene) and AAV2-GFP (UNC viral core) was injected into the ACC, and a 90 nl injection of AAV2-FLEX-TdTomato (UNC viral core) was injected in the ipsilateral MThal. Three weeks post-injection, mice were

transcardially perfused as described above and brains were sectioned 50 μm thick in PBS the following day.

## Immunohistochemistry and imaging for retrograde viral cell label

All incubation steps were performed on a shaker and at room temperature unless otherwise stated. Immediately post-slicing, sections were washed with PBS, and then permeabilized for 20 min with 1% Triton X-100 in PBS. Sections were subsequently incubated for 1 hr in blocking solution containing PBS with 1% triton X-100% and 0.5% fish skin gelatin (FSG). Rabbit anti-DsRed (Clontech, Mountain View, CA) and chicken anti-GFP (Invitrogen, Carlsbad, CA) primary antibodies diluted 1:500 in the blocking solution were incubated at 4°C overnight. Secondary antibodies were diluted 1:750 and incubated for 2 hr in the blocking solution. Sections were stained with 300 nM DAPI for 30 min to label nuclei, then mounted on microscope slides and embedded in an aqueous-based mounting solution (Fluoromount, Sigma Aldrich, Saint Louis MO). Imaging conditions are similar to retrograde bead labeling experiments. Red and green axons in the DMS were imaged at 63x using a Zeiss LSM 880 with Airyscan on a single tile with 0.44 μm optical section thickness.

## PV/DOR In Situ hybridization

Advanced Cell Diagnostics RNAscope Technology (ACD Bioscience, Newark, CA) was used to quantify cells containing *Oprd1* and *Pvalb* mRNA. Briefly, wild-type mice (five to eight weeks old) were deeply anesthetized with ketamine-xylazine and perfused transcardially with 0.1 M PBS, followed by 4% formaldehyde solution in phosphate buffer (PB). Brain was dissected, cryoprotected in 30% sucrose overnight and then frozen in OCT. Frozen tissue was sectioned at 20 μm, transferred onto Superfrost Plus slides and kept at −80°C. Tissue was thawed from −80°C, washed with PBS at room temperature and subsequently processed according to the manufacturer's protocol. We first pretreated the tissue with solutions from the pretreatment kit to permeabilize the tissue, and then incubated with protease for 30 min and the hybridization probes for another 2 hr at 40°C (*Wang et al., 2018*).

## PV/DOR Immunohistochemistry

A previously described immunostaining protocol was employed (*Bardoni et al., 2014*; *Scherrer et al., 2009*). Briefly, five to eight week-old mice were deeply anesthetized with ketamine-xylazine and perfused transcardially with 0.1 M PBS, followed by 4% formaldehyde solution in 0.1 M PB. The brain was dissected, post-fixed for 4 hr at 4°C, and cryoprotected overnight in 30% sucrose in PBS. Frozen brain tissue was then sectioned at 40 μm and incubated with a 5% NDST blocking solution (0.3% Triton X-100 solution in 0.1 M PBS plus with 5% normal donkey serum) for at least 1 hr. The primary antibody was diluted in the same solution, and incubated with brain sections overnight at 4°C. After washing the primary antibody three times for 5 min with 0.3% Triton X-100 solution in 0.1 M PBS, sections were incubated with secondary antibody solution in 1% NDST solution at room temperature for 2 hr. Sections were then mounted on microscope slides with Fluoromount (Southern Biotech, Birmingham, AL) after washing with PB for three times for 5 min. Images were acquired with a confocal microscope (Leica DM2500, Wetzlar, Germany). The following primary antibodies were used: anti-GFP: Abcam (chicken; 1:1,000); anti-Parvalbumin: Swant (goat; 1:1,000).

## Quantification and statistical analysis

For electrophysiology experiments, to avoid observer-bias all data was quantified using automated custom-written analysis software (*Jongbloets et al., 2019*) followed by manual confirmation. For insitu hybridization experiments, signals were manually counted, and cells displaying five or more labeled dots in their cytoplasm were considered positive. All experiments involving conditions (baseline, agonist, antagonist) were first tested with a Skillings-Mack test for significant changes in any of the conditions. Only when a significant change was reported ($\alpha = 0.05$), three Wilcoxon signed-rank tests between combination of conditions was performed, unless stated otherwise. The Skillings-mack test is a non-parametric test, which allows for missing data points (unbalanced design). Dual-channelrhodopsin experiments (*Figure 1D* and Figure 1—figure supplement 1h S2H) required a linear mixed model (LMM) since a 3-way ANOVA could not be performed due to unbalanced design. LMM accounted for correlation of measurement within individual cells (random effects). The fixed effects

in the LMM were opioid type (mu vs. delta), source (MThal vs. ACC/PFC), condition (baseline vs. agonist vs. antagonist), as well as all interactions referred in the text as two-way (opioid type x source, opioid type x condition, source x condition) and three-way (opioid type x source x condition). Due to the paucity of observations (n = 7/8) and high number of fixed effects, we relaxed the type I error for the LMM to detect significant trends in interactions ($\alpha$ = 0.15). *Post-hoc* analysis of *a priori* hypothesis for specific comparisons were performed using linear combinations based on the LMM ($\alpha$ = 0.05). All source data and related custom software for quantification and statistical analysis are available at GitLab (*Jongbloets et al., 2019*; copy archived at https://github.com/elifesciences-publications/opi_syn_circuit). The number of experiments performed with independent mice (N) and recorded neurons or counted slices, in case of in situ hybridization or immunohistochemistry, (n) is indicated in the legends. Error bars represent standard error of the mean.

## Acknowledgements

We would like to thank Dr. John Williams for providing guidance and financial support to WTB (NIH R01DA08136), and Dr. Emmeke Aarts and Sheila Markwardt MPH for providing guidance and comments on the statistical analyses. We thank Drs. John Williams, Haining Zhong, and Brooks Robinson for comments to the manuscript. We thank Sweta Adhikary for assistance with retrobead cell counting. This work was supported by NIH grants R01DA042779 (WTB), R01DA044481 (GS), R01NS106301 (GS), R01NS081071 (TM) and U01NS094247 (TM), R01NS104944 (TM), the Department of Defense Neurosensory Award MR130053 (GS) and the New York Stem Cell Foundation (GS). GS is a New York Stem Cell Foundation - Robertson Investigator.

## Additional information

### Funding

| Funder | Grant reference number | Author |
|--------|------------------------|--------|
| National Institute of Neurological Disorders and Stroke | R01NS081071 | Tianyi Mao |
| New York Stem Cell Foundation | | Gregory Scherrer |
| National Institute on Drug Abuse | R01DA042779 | William T Birdsong |
| National Institute on Drug Abuse | R01DA044481 | Gregory Scherrer |
| National Institute on Drug Abuse | R01NS106301 | Gregory Scherrer |
| National Institute of Neurological Disorders and Stroke | R01NS104944 | Tianyi Mao |
| National Institute of Neurological Disorders and Stroke | U01NS094247 | Tianyi Mao |

The funders had no role in study design, data collection and interpretation, or the decision to submit the work for publication.

### Author contributions

William T Birdsong, Conceptualization, Resources, Formal analysis, Supervision, Funding acquisition, Investigation, Visualization, Methodology, Writing—original draft, Writing—review and editing; Bart C Jongbloets, Conceptualization, Data curation, Software, Formal analysis, Validation, Investigation, Visualization, Methodology, Writing—original draft, Writing—review and editing; Kim A Engeln, Dong Wang, Investigation, Visualization, Methodology, Writing—review and editing; Grégory Scherrer, Resources, Supervision, Funding acquisition, Methodology, Writing—review and editing; Tianyi Mao, Conceptualization, Resources, Supervision, Funding acquisition, Validation, Methodology, Writing—original draft, Writing—review and editing

Author ORCIDs
Bart C Jongbloets (iD) https://orcid.org/0000-0003-4799-333X
Tianyi Mao (iD) https://orcid.org/0000-0002-3532-8319

Ethics
Animal experimentation: All procedures were approved by Oregon Health & Science University Institutional Animal Care and Use Committee (IACUC) and all experiments were performed strictly according the approved protocols. IACUC protocol IP00000955, and Institutional Biosafety Committee protocol IBC-10-40.

Decision letter and Author response
Decision letter https://doi.org/10.7554/eLife.45146.018
Author response https://doi.org/10.7554/eLife.45146.019

## Additional files

### Supplementary files
• Transparent reporting form
DOI: https://doi.org/10.7554/eLife.45146.016

### Data availability
All data generated or analyzed during this study are included in the manuscript and supporting files. All code and data are deposited in https://gitlab.com/maolab/opi_syn_circuit (copy archived at https://github.com/elifesciences-publications/opi_syn_circuit).

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
