## [Decision Letter]

Thank you for submitting your article "Synapse-specific Opioid Modulation of Thalamo-cortico-striatal Circuits" for consideration by *eLife*. Your article has been reviewed by Ronald Calabrese as the Senior Editor, a Reviewing Editor, and three reviewers. The following individuals involved in review of your submission have agreed to reveal their identity: Christopher Ford (Reviewer #1); Carlos Paladini (Reviewer #2).

The reviewers have discussed the reviews with one another and the Reviewing Editor has drafted this decision to help you prepare a revised submission.

Summary:

This study examines how opioid receptors modulate synaptic transmission in thalamo-cortico-striatal circuits. Opioids strongly regulates these circuits and determining exactly where in these circuits these receptors act to gate thalamo-cortical activity into the striatum is an important question. The authors used optogenetics (two color ChR2 chrimson/chronos) in brain slices to thoroughly identify where and how mu and δ opioid receptor alter circuit activity in these regions. They show that MORs but not DORs strongly inhibit medial thalamic glutamatergic inputs to the DMS and ACC, and that a proportion of mThal neurons that project to the ACC also project to the DMS. Within this circuit DORs are expressed primarily on PV interneurons in the ACC where they function to regulate feed-forward inhibition into the ACC. Finally, they show that MORs inhibit thalamo-cortical-striatal circuits while DORs alternatively disinhibit these circuits. This work advances our understanding of how opioids mechanistically can alter circuit activity that likely underlies the affective-motivational aspects of pain processing.

Essential revisions:

The following points must be addressed before final acceptance of the manuscript. Most of these concerns can be addressed with additional analysis, modifications of the figures and extension of the Discussion section. For your understanding of these essential revisions, the full reviews are appended below.

1) In Figure 3E,F, it is unclear what proportion of MSNs exhibited an EPSC when AAV2-retro-Cre was injected into the AAC and AAV-DiO-ChR2 was injected into the MThal. Further describing the proportion of cells in the DMS exhibited EPSCs would help show the distribution of cells that receive input from mThal cells that also project to the ACC. A scatter plot of the responders/non-responders of MSNs that exhibited EPSCs would be informative.

2) There is concern about the lack of reversal (due to no application of antagonist) or application of antagonist that does not reverse the opioid inhibition of the synaptic response/firing of the cell. In several experiments neurons are included in the analysis where the opioid inhibition is not reversed by the antagonist (e.g. Figure 4C,F; Figure 4—figure supplement 1C,F, Figure 5C). There also appear to be cells that have a data point for the agonist but without a connected data point for the antagonist. For example, Figure 4—figure supplement 1, Figure 4C,F, Figure 5. Of particular concern is the data shown in figure 4f which is pivotal to the interpretation of the data presented in this study. The effect of DPDPE on the IPSC seems very variable. In some cells the agonist inhibited the response and this was sometimes reversed, sometimes no antagonist response is shown (for largest inhibition) and in some cases the antagonist produced a large overshoot in the size of the synaptic current. There only appears to be one cell where δ receptor activation reduces the size of the IPSC and this is reversed to baseline by the antagonist. How could these changes in synaptic responses be due to activation of the relevant opioid receptor if they were not reversed by the relevant antagonist? If there is a biological basis for it, rather than experimental, it would be interesting to see it discussed.

3) The authors suggest that overall activation of the striatum will determine whether the person experiences the affective aspect of pain (Introduction). Is there evidence that activation of the striatum, regardless of whether this comes from the thalamus or cortex, is aversive? Or do the different inputs to the striatum result in a different experience of the aversive component of pain?

4) It is surprising that, given the substantial circuit differences from D1 and D2 spiny projection neuron activation, that D1 and D2 SPNs were not identified. Although the differential effect of MOR agonist alleviates this concern, it is nonetheless preferable to have an independent measure that each laser activates distinct pathways. When activating the 470 laser and the 625 laser simultaneously, the currents should sum linearly if they are from distinct inputs.

5) Does Figure 6B show action potentials? These must be escaped action currents. Nevertheless, they appear to have half-widths of 3 ms. Using escaped action currents as a measure of firing rate may be affected by the "voltage command" even when not broken in to the whole-cell mode.

6) Do the specific MThal inputs to ACC synapse on the ACC projections to DMS?

7) Although different opioids may have differing effects, it is difficult to discern what the endogenous effect of opioids may be on the circuit. MOR activation on MThal to ACC pyramidal pathway causes a decrease in EPSC amplitude and a decrease in IPSC amplitude. Meanwhile, DOR activation causes a decrease in IPSC amplitude in ACC pyramidal neurons. What is the net effect on firing?

*Reviewer #1:*

In the present manuscript Birdsong and Jongloets et al., examine how opioid receptors modulate synaptic transmission in thalamo-cortico-striatal circuits. Over a series of careful experiments, the authors use optogenetics in brain slices to thoroughly identify where and how mu and δ opioid receptor alter circuit activity in these regions. As opioids are well known to strongly regulate these circuits determining exactly where in these circuits these receptors act to gate thalamo-cortical activity into the striatum is a significant advance. The authors make good use of two color ChR2 (chrimson/chronos) to determine that MORs but not DORs strongly inhibit medial thalamic glutamatergic inputs to the DMS and ACC, and that a proportion of mThal neurons that project to the ACC also project to the DMS. They go on to show that within this circuit DORs are expressed primarily on PV interneurons in the ACC where they function to regulate feed-foreward inhibition into the ACC. Finally, using the clever combination of cell attached recordings and dual recordings in the ACC and DMS they show that MORs inhibit thalamo-cortical-striatal circuits while DORs alternatively disinhibit these circuits.

The experiments and controls have been performed well and carefully analyzed/interpreted. Based on the work provided in the manuscript and supplemental figures I believe that the conclusions are well justified and no further experiments are required. The exciting aspect of this work is that marks a significant advance in furthering our understanding of how opioids mechanistically can alter circuit activity that likely underlies the affective-motivational aspects of pain processing. Together, with the carefully performed experiments and well written manuscript make it appropriate for publication in *eLife*.

My only suggestion relates to the presentation of data in Figure 3. It was unclear what proportion of MSNs exhibited an EPSC when AAV2-retro-Cre was injected into the AAC and AAV-DiO-ChR2 was injected into the MThal (Figure 3E,F)? A scatter plot of the responders/non-responders of MSNs that exhibited EPSCs would be informative. As EPSCs are much smaller than previous figures from mThal injections it seems as though a relatively small numbers of cells projecting to the ACC also project to the DMS. Further describing the proportion of cells in the DMS exhibited EPSCs would help show the distribution of cells that receive input from mThal cells that also project to the ACC.

*Reviewer #2:*

The manuscript by Birdsong et al., investigates the influence of opioid receptor activation on separate afferents to spiny projection neurons in the striatum. The anterior cingulate cortex and the medial thalamus projections were individually stimulated by optogenetic activation of either Chrimson or ChR2. The mThal to Str pathway was sensitive to mu receptor agonism whereas the ACC to Str pathway was insensitive to both MOR and DOR activation. However, the mThal to ACC projection was sensitive to MOR activation. This creates a circuit whereby mThal afferents synapsing directly on ACC pyramidal neurons are sensitive to MOR while inhibitory interneurons in the ACC are sensitive to DOR. This is an exciting finding as it shows the differential effect that DOR and MOR can have on spiny projection neurons in the dorsal medial striatum. This can have translational applications in the use of opioids with higher affinities for either DORs or MORs.

Although different opioids may have differing effects, it is difficult to discern what the endogenous effect of opioids may be on the circuit. MOR activation on MThal to ACC pyramidal pathway causes a decrease in EPSC amplitude and a decrease in IPSC amplitude. Meanwhile, DOR activation causes a decrease in IPSC amplitude in ACC pyramidal neurons. What is the net effect on firing?

It is surprising that, given the substantial circuit differences from D1 and D2 spiny projection neuron activation, that D1 and D2 SPNs were not identified.

Although the differential effect of MOR agonist alleviates this concern, it is nonetheless preferable to have an independent measure that each laser activates distinct pathways. When activating the 470 laser and the 625 laser simultaneously, the currents should summate linearly if they are from distinct inputs.

*Reviewer #3:*

This paper addresses how we can understand the overall effect of different opioid receptor activity acting at multiple sites within important brain circuits. The optogenetic stimulation of multi-synaptic responses in striatal neurons allows direct testing of these circuit effects rather than inference from responses at multiple sites which is a significant advance.

I have a few concerns about the experiments included in some experiments and data that either wasn't included or experiments not performed.

I have some concerns about the lack of reversal (due to no application of antagonist) or application of antagonist that does not reverse the opioid inhibition of the synaptic response/firing of the cell. For some experiments the data is very clear, good inhibition with an agonist and full reversal with an antagonist, for example at the mthal-str synapses in figure 2. However, in several other experiments neurons are included in the analysis where the opioid inhibition is not reversed by the antagonist. For example, Figure 4C,F; Figure 4—figure supplement 1C,F, Figure 5C. There also appear to be cells that have a data point for the agonist but without a connected data point for the antagonist. For example, Figure 4—figure supplement 1, Figure 4C,F, Figure 5. Of particular concern is the data shown in figure 4f which is pivotal to the interpretation of the data presented in this study. The effect of DPDPE on the IPSC seems very variable. In some cells the agonist inhibited the response and this was sometimes reversed, sometimes no antagonist response is shown (for largest inhibition) and in some cases the antagonist produced a large overshoot in the size of the synaptic. There only appears to be one cell where δ receptor activation reduces the size of the IPSC and this is reversed to baseline by the antagonist. How can you be confident that the changes in synaptic responses were due to activation of the relevant opioid receptor if they were not reversed by the relevant antagonist? This needs to be resolved and if there is a biological basis for it, rather than experimental, it would be interesting to see it discussed.

In other experiments antagonist reversals are not shown. For example, data depicted in Figure 6 and Figure 7. This reduces the confidence in this data.

It would also have been interesting to see the effect of MOR activation in the experiment using light activation in the ACC presented in Figure 7. This would have further tested the model of how opioids are acting in this circuit.

The authors suggest that overall activation of the striatum will determine whether the person experiences the affective aspect of pain (Introduction). Is there evidence that activation of the striatum, regardless of whether this come from the thalamus or cortex, is aversive? Or do the different inputs to the striatum result in a different experience of the aversive component of pain?

---

## [Author Response]

Essential revisions:The following points must be addressed before final acceptance of the manuscript. Most of these concerns can be addressed with additional analysis, modifications of the figures and extension of the discussion. For your understanding of these essential revisions, the full reviews are appended below.1) In Figure 3E,F, it is unclear what proportion of MSNs exhibited an EPSC when AAV2-retro-Cre was injected into the AAC and AAV-DiO-ChR2 was injected into the MThal. Further describing the proportion of cells in the DMS exhibited EPSCs would help show the distribution of cells that receive input from mThal cells that also project to the ACC. A scatter plot of the responders/non-responders of MSNs that exhibited EPSCs would be informative.

We agree with the reviewers that the question of how frequently ACC-projecting MThal inputs synapse to the MSNs in the DMS is an interesting one but might not be possible to address with the limitation of existing reagents and methods. This is mainly due to two factors, both of which affect the interpretation of non-responders. (1) The existing retrograde viruses, including rAAV2-retro we used here, have been improved in terms of label efficiency but are far from 100%. Therefore, we cannot virally infect all axons within the ACC that arise from the MThal by our viral combination of rAAV2-retro expressing Cre-recombinase in the ACC and AAV2 expressing DIO-ChR2-EYFP in the MThal. As a result, if an MSN does not exhibit an EPSC upon light stimulation, it does not necessarily mean that it is not receiving ACC-projecting MThal inputs. (2). Whether an EPSC, thus a synaptic connection, can be detected also depends on the expression level of ChR2, which can vary across animals, and is a known caveat of the optogenetic tools. The smaller EPSCs we observed in these experiments (Figure 3F) relative to other experiments (Figure 1C and Figure 2B) is likely due to the low efficiency of the retrograde Cre-recombinase expression in the MThal cells which resulted in much lower expression level of ChR2 in the MThal compared to that expression observed in other experiments.

In summary, given the limitation of the current technology and intersectional nature of this experiment, in our system, the lack of EPSCs does not lead to the conclusion of lack of bona fide synaptic connections. Therefore, we feel that in this case, the scatter plot will not address the question the reviewer is interested in. In the meantime, we now emphasize in the main text that what we can claim is the existence of the ACC-projecting MThal neurons which also project to the MSN in the DMS, and this projection is sensitive to enkephalin (subsection “Thalamostriatal and thalamocortical projections can arise from the same medial thalamic neuronal population”).

2) There is concern about the lack of reversal (due to no application of antagonist) or application of antagonist that does not reverse the opioid inhibition of the synaptic response/firing of the cell. In several experiments neurons are included in the analysis where the opioid inhibition is not reversed by the antagonist (e.g. figure 4C,F; Figure 4—figure supplement 1C,F, Figure 5C). There also appear to be cells that have a data point for the agonist but without a connected data point for the antagonist. For example, Figure 4—figure supplement 1, Figure 4C,F, Figure 5. Of particular concern is the data shown in Figure 4F which is pivotal to the interpretation of the data presented in this study. The effect of DPDPE on the IPSC seems very variable. In some cells the agonist inhibited the response and this was sometimes reversed, sometimes no antagonist response is shown (for largest inhibition) and in some cases the antagonist produced a large overshoot in the size of the synaptic current. There only appears to be one cell where δ receptor activation reduces the size of the IPSC and this is reversed to baseline by the antagonist. How could these changes in synaptic responses be due to activation of the relevant opioid receptor if they were not reversed by the relevant antagonist? If there is a biological basis for it, rather than experimental, it would be interesting to see it discussed.

We believe that the variability in effect of DPDPE on the IPSCs is a mix of experimental limitations in this biological system. First, the IPSCs are generally large and disynaptic or polysynaptic meaning small changes in excitability can have large effects. If the IPSCs are polysynaptic, inhibiting the PV cells would lead to disinhibition of other pyramidal cells which may in turn excite more PV cells as a negative feedback. Because a small number of PV cells densely innervate pyramidal cells, adding or losing the input of a single PV cell can have a large effect on IPSC amplitude particularly when the frequency of optical stimulation is low and the number of trials that are averaged is limited.

We have established standards for recording quality and health of cells. In some experiments, cell health or recording criteria did not hold long enough for a full set of pharmacology experiments, e.g., only agonists were added and no antagonists were added. We still include those cells with the part that all criteria were met. That is why some data points did not have antagonist data (these data points lack a gray dashed line). For this reason, we were very careful with our statistical analyses and utilized steps to specifically deal with the unmatched data points. As stated in subsection “Quantification and Statistical Analysis”, “The Skillings-mack test is a non-parametric test, which allows for missing data points (unbalanced design).”

The lack of reversal in some examples may be due to incomplete reversal as a result of competition between DPDPE and naltrindole. Additionally, desensitization or rundown of the channelrhodopsin currents in the MThal to PV synapses may also contribute to the non-full reversal with antagonists. In combination with the notion above that small changes in the EPSP amplitude evoked in putative PV neurons could bring the PV cells below action potential threshold, a small change in channel rundown can have a major effect in this experimental setup.

In addition, these experiments involve stepping the membrane potential between -60 mV and +5 mV. There was potentially some contamination of the IPSCs with EPSCs as a small inward current is noticeable in the example traces in Figure 4E. Small drift in the actual holding potential or changes in space clamping over the experiment could also result in slight changes of the relative contributions of EPSCs and IPSCs, and thus the peak amplitude of the IPSCs measured.

In Figure 5, multiple possibilities also exist for some non-full reversal effects. We believe that the reversal of DPDPE inhibition by naltrindole is slow, and therefore may not have been completely reversed in a timeframe relevant for these experiments. This could be due to kinetics of competition binding, or the inhibition may be mediated in part by kinase activity which can have effects after G-protein signaling is terminated. In addition, some other forms of plasticity could be induced by DOR activation which would be slow to reverse on the timescale of these experiments.

We do want to experimentally address whether the issue is due to non-specific effects of the pharmacological reagents we used. The rationale is that one potential possibility of the partial irreversibility is because DPDPE may modulate synaptic transmission via DOR-independent pathway, therefore DOR antagonist naltrindole would not able to reverse the DPDPE-mediated modulation of IPSCs. Therefore, we have performed an additional experiment which is to repeat the experiments shown in Figure 4D-F, except that we applied DOR-antagonist naltrindole first, followed by DOR-agonist DPDPE. AAV2-CAG-ChR2(H134R)-TdTomato virus was injected in the MThal, and feed-forward inhibitory inputs to L5 pyramidal neurons the ACC were evaluated (Figure 4—figure supplement 3). We recorded optically-evoked IPSCs (oIPSCs), and first perfused DOR-antagonist naltrindole followed by DOR-agonist DPDPE. Naltrindole pre-treatment prevented the DPDPE-mediated modulation of oIPSC, as observed in Figure 4 and Figure 5, (Figure 4—figure supplement 3), indicating that DPDPE modulates GABAergic neurotransmission via a DOR-dependent pathway.

3) The authors suggest that overall activation of the striatum will determine whether the person experiences the affective aspect of pain (Introduction). Is there evidence that activation of the striatum, regardless of whether this comes from the thalamus or cortex, is aversive? Or do the different inputs to the striatum result in a different experience of the aversive component of pain?

We thank the reviewers for pointing this out. Our original rationale was that since both increased ACC and medial thalamic activities are known to be associated with pain perception (e.g., Casey et al., 1994; Davis et al., 1997; Peyron et al., 1999, Peyron et al., 2000), and these two regions most strongly converge in the DMMS (Hunnicutt et al., 2016), we predicted that decreased activity of these terminals in the DMS would be involved in pain relief, though this has not been directly demonstrated to our knowledge. We have now re-worded this part to reflect that MOR and DOR activation is predicted to modulate pain responses in opposite ways (Introduction, “Thus, MOR and DOR activation are predicted to play opposite roles in pain processing mediated by this thalamo-cortico-striatal circuit”), and we think this is better reflective of our data and current knowledge.

4) It is surprising that, given the substantial circuit differences from D1 and D2 spiny projection neuron activation, that D1 and D2 SPNs were not identified. Although the differential effect of MOR agonist alleviates this concern, it is nonetheless preferable to have an independent measure that each laser activates distinct pathways. When activating the 470 laser and the 625 laser simultaneously, the currents should sum linearly if they are from distinct inputs.

We agree with the reviewers that investigating cell type-specific effects onto D1 and D2 MSNs would be interesting. So far, under our experimental conditions, we have not observed obvious heterogeneous responses, at least not a bi-modal distribution as one might expect if there are major differences in D1 versus D2 responses, of opioid inhibition of MThal inputs to the MSNs, indicating that inputs to both D1 and D2 MSNs might be modulated in a similar fashion (Author response image 1). Previous work comparing the excitatory input kinetics between D1 and D2 MSNs suggest that the risetime of mEPSCs are slower in D2 MSNs (Al-muhtasib et al., 2018). If the DAMGO effect on EPSC amplitude is different between D1 versus D2 MSNs, a correlation is expected between baseline EPSC risetime and DAMGO effect on EPSC amplitude in our data. Under current conditions, no significant correlation between DAMGO effect on EPSC amplitude and baseline EPSC risetime obtained from all MSNs is observed (Author response image 1). Furthermore, comparison of two putative MSN subpopulations, defined using a cluster analysis among all recorded MSNs based on the baseline EPSC kinetics (Figure 4—figure supplement 3C), yielded non-significantly different distributions of DAMGO effect on EPSC (Figure 4—figure supplement 3D). Together, this suggests that so far, there is no evidence that observed mu-opioid mediated suppression of thalamostriatal inputs is dependent on the identity of the recorded MSN. We think the systematic investigation of cell-type specific response is beyond the scope of current manuscript but an interesting question for future investigation. We have now added a notion that we have not observed any evidence for opioids’ modulation of the thalamic inputs that suggest potential D1 and D2 MSN specific effects (subsection “Opioid modulation of the thalamo-cortico-striatal circuitry”).

**Author response image 1. respfig1:** No clear bi-modal or two population effects detected in MOR-mediated suppression of thalamostriatal inputs to the MSNs in the DMS. (**a**) Histogram of cell distribution based on effects of mu-opioid agonist DAMGO on optically-evoked excitatory postsynaptic current (EPSC) amplitude recorded in the MSNs; n = 40 neurons from 25 mice. The observed DAMGO effects had a non-normal distribution (Shapiro-Wilk normality test, *W* = 0.8219, p < 0.001), but showed no clear bi-modal distribution of two roughly equal populations. (**b**) Plot of the DAMGO effects on baseline oEPSC amplitude recorded from MSNs as a function of risetime of the baseline EPSC. There was no correlation between EPSC risetime and degree of EPSC inhibition by DAMGO. Linear regression model (red dash line), R^2^ = 0.018, F (1, 38) = 0.677, p = 0.416. (c-d) In case two populations exist within the recorded MSNs, we wanted to test whether the effect of DAMGO on baseline EPSC amplitude was significantly different in the two populations defined based on baseline EPSC parameters. (**c**) Agglomerative hierarchical cluster analysis based on Pearson correlation between baseline EPSC parameters (columns), and Euclidean distance between recorded MSNs (rows), with heatmap representing centered/standardized values (x-x-σ). Two main clusters were defined (cluster 1: red n = 16 neurons/11 mice, cluster 2: blue n = 23 neurons/17 mice) based on baseline EPSC parameters, representing putative subpopulations among the recorded MSNs. (**d**) Histograms of cell distributions based on DAMGO effects on baseline recorded from the MSNs defined in cluster 1 or 2, red or blue, respectively. Mann-Whitney U test, *U* = 204, p = 0.751.

Regarding the dual channelrhodopsin activation experiments, we addressed the potential cross-activation of Chrimson with 470 nm excitation in Figure 1—figure supplement 1D-E by expressing each channelrhodopsin variant separately in individual animals, and stimulated with both the 470 nm and 625 nm excitation. From these data the degree of cross-activation was estimated to be very low (approximately 4% for Chrimson by 470 nm, and < 1% for ChR2 and CsChR by 625 nm). We feel that the cross-activation data together with the data demonstrating a clear inhibition of the 470 nm-evoked currents by DAMGO with no effects on the 625 nm-evoked currents should alleviate the concern of unintended activation with 625 nm light. To further verify our results and conclusions that specific opioid receptor types modulate distinct inputs to the striatum we performed independent control experiments as shown in Figure 1—figure supplement 2. In these experiments individual projection origins, i.e. ACC, PFC, AMThal, and MThal, were injected with only one ChR2 variant, CsChR, and subsequently tested their sensitivity to opioid modulation. We confirmed that DAMGO specifically modulated the thalamostriatal projections, and therefore that these projections represent functionally and pharmacologically distinct inputs compared to the corticostriatal projections. Thus, we feel confident that there was very little cross talk and any potential cross-activation did not change any interpretations of the data.

We did not attempt the additivity analysis suggested due to the fact that the activation of Chrimson by 470 nm light is still not 0 (as mentioned above, 4%; Figure 1—figure supplement 1). While the low intensity of 470 nm stimulation we used may not evoke large EPSCs, we cannot exclude the possibility that 470 nm and 625 nm light together lead to non-linear effects for reasons beyond our current understanding of these tools.

5) Does Figure 6B show action potentials? These must be escaped action currents. Nevertheless, they appear to have half-widths of 3 ms. Using escaped action currents as a measure of firing rate may be affected by the "voltage command" even when not broken in to the whole-cell mode.

Figure 6B shows action currents recorded in cell-attached mode. Voltage clamp mode setting on the amplifier was used and the voltage command was set and adjusted when needed during the experiment, to obtain an amplifier current of 0 pA. This approach ensures that the spike activity of the recorded neuron was not influenced by the cell-attached mode settings, since no current is flowing between the cell and electrode (Perkins, 2006). The firing rate therefore should not be affected by the voltage command. In addition, quantification of the action potential widths (width at half-max action current amplitude, Author response image 2) during baseline and agonist applications (DAMGO and DPDPE) were combined, showing that the mean action potential width is 1.36 ms, consistent with the reported action potential widths in L5 pyramidal neurons in the mouse ACC (1.13 ms, Lee et al., 2007) and in L5 pyramidal neurons in the neocortex (2.08 ms, see neuronelectro.org). In conclusion, the recorded currents resemble action potentials with expected kinetics, which remain stable during the recording and therefore, we do not have evidence that ‘voltage command’ in cell-attached mode affect our results.

**Author response image 2. respfig2:** Action potential widths remain stable during cell-attached recording and they are comparable to other studies in the L5 pyramidal neurons on the cortex. Summary data of median full width at half-maximum of action potential currents recorded in L5 pyramidal neurons of the ACC; n = 14 neurons from 5 mice. The agonists (DAMGO or DPDPE) did not change the recorded action potential current width; Mann-Whitney U test *U* = 189, p = 0.5320.

6) Do the specific MThal inputs to ACC synapse on the ACC projections to DMS?

To answer the question, we have performed additional experiments to record from the L5 pyramidal neurons in the ACC that project to the DMS to test whether they receive MThal inputs (Author response image 3). To label the DMS-projecting L5 pyramidal neurons in the ACC (Author response image 3), we injected retrograde beads in the DMS; and to optically stimulate the MThal inputs, we injected AAV2-CAG-ChR2(H134R)-tdTomato in the MThal of the same animal (Author response image 3). From all recorded DMS-projecting L5 pyramidal neurons, ~91.7% (11 out of 12 neurons) received MThal inputs (Author response image 3) with a charge transfer and onset value within the range of EPSCs reported in the main text (Author response image 3, see also Figure 4—figure supplement 2C and Figure 7—figure supplement 1A). These results indicate that indeed, MThal inputs to ACC synapse onto the DMS-projecting neurons in the ACC.

**Author response image 3. respfig3:** The DMS-projecting L5 pyramidal neurons in the ACC receive direct inputs from the MThal. (**a**) Experimental schematic showing retrogradely-transported beads (retrobeads) injection in the DMS to retrogradely-label DMS-projecting pyramidal neurons (Pyr) in the ACC, and AAV2/1-CAG-ChR2(H134R)-tdTomato injection in the MThal to allow for optical stimulation of specific thalamic inputs to the ACC. (**b**) Representative brightfield (left panel) and fluorescent (center panel) images of an ACC layer 5 (L5) with retrobeads present in the soma (merge, right panel). (**c**) Averaged optically-evoked excitatory postsynaptic currents (EPSCs, average of 12 single trials) of all individual bead-positive L5 pyramidal neurons recorded in the DMS/MThal injected mice (n = 12 neurons from 4 mice). 11 out of 12 recorded cells (black traces) received inputs from the MThal, defined as peak amplitude > (mean baseline + 5 x standard deviation). Grey trace: non-responding L5 pyramidal neuron. Blue bar: 1 ms of 470 nm light stimulation. (**d**) Quantification of the EPSC charge transfer of all recorded EPSCs as shown in B. (**e**) Quantification of EPSC onset time of all recorded EPSCs as shown in B. Box plot with median, first and third quartile and whiskers represent 1.5 times interquartile range.

7) Although different opioids may have differing effects, it is difficult to discern what the endogenous effect of opioids may be on the circuit. MOR activation on MThal to ACC pyramidal pathway causes a decrease in EPSC amplitude and a decrease in IPSC amplitude. Meanwhile, DOR activation causes a decrease in IPSC amplitude in ACC pyramidal neurons. What is the net effect on firing?

We agree with the reviewers that the endogenous effect of opioids is an import question and a future direction to take. However, addressing the endogenous effects of opioid receptor activation is currently difficult and is beyond the scope of the current work. The main obstacle for addressing this question is the ability to reliably release and quantify the release of endogenous opioids, e.g., enkephalin or dynorphin in mouse brain slice. We did attempt to trigger release of enkephalin in transgenic mouse lines, which expressed channelrhodopsin in putative enkephalin-releasing cells (*PENK-IRES2-Cre;Ai32* mice). Unfortunately, it was difficult to establish the reliable stimulation protocols for enkephalin release due to lack of reliable ways to measure enkephalin release. For the same reason, it is difficult to pharmacologically address the net effects on firing without knowing whether/how the endogenous opioid receptor is activated. These pharmacological agents, e.g. enkephalins, endorphins and dynorphins, have differential affinities for mu, δ and kappa opioid receptors, though the affinity differences can be minimal between mu and δ receptors (e.g. Raynor et al., 1994). Differences in affinity, the sites of action, and the concentration of endogenous agonists will together determine the net effect of firing in the ACC. Based on our findings, our prediction is that at modest concentrations of endogenous opioid receptor ligands with highest affinities for MORs preferably reduce ACC firing, and therefore suppress both direct thalamostriatal and the indirect thalamo-cortico-striatal pathways, whereas those endogenous ligands with highest affinities for DORs, at relatively selective concentrations increase ACC firing. Because IPSCs are sensitive to both mu and δ agonists, low to moderate concentrations of non-selective agonists would be predicted to impact feed-forward inhibition more profoundly and have a net excitatory effect, while higher concentration would greatly diminish EPSCs producing a net inhibitory effect. The future work that measures the endogenous ligand release sites and concentrations will provide key information for answering these questions. Since the direct testing of these predictions is not feasible with our current means, and the related topic might be work of its own in the future, we have not included major discussions but have added a short one to this point (subsection “Opposing effects of opioid subtypes on circuitry modulation”).